# Cognitive cascades: How to model (and potentially counter) the spread of fake news

**Nicholas Rabb** [1]*, **Lenore Cowen**[1], **Jan P. de Ruiter**[1,2], **Matthias Scheutz**[1]

**1** Department of Computer Science, Tufts University, Medford, Massachusetts, United States of America,
**2** Department of Psychology, Tufts University, Medford, Massachusetts, United States of America

* nicholas.rabb@tufts.edu

**Data Availability Statement:** All data is found within the paper and its Supporting information files. In addition, we provide all code and documentation at: https://github.com/RickNabb/cognitive-contagion.

## Abstract

Understanding the spread of false or dangerous beliefs—often called misinformation or disinformation—through a population has never seemed so urgent. Network science researchers have often taken a page from epidemiologists, and modeled the spread of false beliefs as similar to how a disease spreads through a social network. However, absent from those disease-inspired models is an internal model of an individual's set of current beliefs, where cognitive science has increasingly documented how the interaction between mental models and incoming messages seems to be crucially important for their adoption or rejection. Some computational social science modelers analyze agent-based models where individuals do have simulated cognition, but they often lack the strengths of network science, namely in empirically-driven network structures. We introduce a *cognitive cascade* model that combines a network science belief cascade approach with an internal cognitive model of the individual agents as in opinion diffusion models as a *public opinion diffusion* (POD) model, adding media institutions as agents which begin opinion cascades. We show that the model, even with a very simplistic belief function to capture cognitive effects cited in disinformation study (dissonance and exposure), adds expressive power over existing cascade models. We conduct an analysis of the cognitive cascade model with our simple cognitive function across various graph topologies and institutional messaging patterns. We argue from our results that population-level aggregate outcomes of the model qualitatively match what has been reported in COVID-related public opinion polls, and that the model dynamics lend insights as to how to address the spread of problematic beliefs. The overall model sets up a framework with which social science misinformation researchers and computational opinion diffusion modelers can join forces to understand, and hopefully learn how to best counter, the spread of disinformation and "alternative facts."

## Introduction

Understanding the spread of false or dangerous beliefs through a population has never seemed so urgent. In our modern, highly networked world, societies have been grappling with widespread belief in conspiracies [1–5], increased political polarization [6–9], and distrust in

**Funding:** We thank the Tufts Data Intensive Studies Center (DISC) and National Science Foundation grant NSF-NRT 2021874, for supporting this research. LC and NR also thank NSF CCF-1934553 for additional support.

**Competing interests:** The authors have declared that no competing interests exist.

scientific findings [10, 11]. Some of the most prominent in our times are conspiracies surrounding COVID-19, and starkly polarized distributions of beliefs regarding scientifically-motivated safety measures.

Throughout the course of the pandemic, much effort has been spent trying to understand why, in the face of a global pandemic, so many believed COVID-19 was a hoax, targeted political attack, caused by 5G cell towers, or that it was simply not dangerous and did not justify wearing a protective mask [1, 3–5, 10, 12–14]. Understanding the spread of misinformation requires a way of modeling and understanding both how this misinformation spreads in a population, and also why some individuals are more or less vulnerable.

This paper applies a class of models which capture the spread of ideas, innovations, culture, and more [15] to the spread of misinformation—including components from both social network science and cognitive science. While social network science and cognitive science have both sought to contribute to the understanding of the mechanisms that govern individuals' acquisition and updates of beliefs, each has traditionally focused on different pieces of this puzzle. Social network science has provided interesting insights by applying techniques originally developed to model the spread of disease to modeling the spread of misinformation [2, 16–19]. Modern psychological and cognitive science has focused on the relationship of suggested beliefs to an individual's current set of beliefs, and how this influences an individual's likelihood of updating their beliefs in the face of confirmatory or contradictory information [3, 13, 14, 20–23]. We call our models *cognitive cascade models*—those that adopt a cascading network-based diffusion model from social network science [2, 24, 25], but include a more individually differentiated model of belief update and adoption that is informed by cognitive science as in cognitive contagion models [26–28]. We show that even very simple versions of cognitive cascades result in interesting network dynamics that seem to represent some of the real-world phenomena that were seen in pandemic misinformation.

Misinformation beliefs, once adopted, are so difficult to extricate that over the course of 2020, after forming an initial opinion, the proportion of those who did not believe in the virus hardly changed [29]. In the U.S., reports circulated of nurses in states with few regulations like South Dakota describing patients who would be dying of COVID and refusing to believe they had it. One nurse was quoted saying, "They tell you there must be another reason they are sick. They call you names and ask why you have to wear all that 'stuff' because they don't have COVID and it's not real" [30].

Modern psychological and cognitive science have made a substantive contribution by making a clear distinction between beliefs that update in accordance to the evidence one receives, and others which persist despite clear, logical contrary evidence [22, 31, 32]. This is the distinction that appears key to understanding mechanisms governing belief in misinformation. In fact, there is evidence that those who engage in manufacturing misinformation exploit this research to make their messages more potent. Wiley [33] has recently revealed that some polarized, partisan beliefs and conspiracies have been *designed* to persist despite contrary evidence.

While the study of individual beliefs has recently been advancing, attempting to determine how beliefs, true or otherwise, spread through an entire population adds yet another layer of complexity. Sociologists and political theorists have long studied *public opinion*: the theoretical mechanisms by which populations come to certain beliefs, which are notoriously difficult to verify empirically [34–36]. However, with the advent of social network research, scholars are moving toward that goal, empirically studying how information cascades through groups [2, 24, 37–39], leading to the wide-scale adoption of certain beliefs [40], and motivating theoretical models of networked opinion dynamics [25, 41–45] based off of classic sociological theories [46–48]. Other lines of research develop algorithms on top of those networked opinion models

which optimally manipulate network-wide measures like polarization [49], opinion difference [50], or susceptibility to persuasion [51]. This work is complemented by media studies which theorize about and test mechanisms behind polarization and selective exposure stemming from dissonance pressures [52–57], as well as the role of large media institutions in shaping public opinion [1, 5, 58, 59]. Our work seeks to take one further step: understanding how an integrated model combining individual cognitive belief models with social network and media dynamics might be employed to study how public opinion shifts.

Our work applies a class of *Agent-Based Models* (ABMs), called *Agent Based Social Systems* (ABSS) [60, 61]—more specifically, *social contagion* models [25]—to the study of network cascades, which measure the number people a given story spreads to via sharing [2]. ABM is a powerful modeling paradigm that has both successes and future potential in a variety of areas, including animal behavior [46, 62–65], social sciences [66, 67], and, notably, opinion dynamics [25, 44, 68, 69].

By combining social contagion paradigms with some of the cognitive literature regarding misinformation, we propose a *cognitive cascade* model that captures the spread of identity-related beliefs. This model is tuned to capture, on the individual agent-level, two major effects cited in misinformation literature: dissonance [20], and exposure [10, 70], capturing what we call *defensive cognitive contagion* (DCC).

We then situate agents following different classes of contagion rules in a cascade model that includes *institutional* agents who begin the cascades by injecting messages into the network. Here, we define *cascades* as the spread of messages via sharing through the population, initiated by institutional agents. Where we say *contagion*, we mean the spread of a belief between two agents, as opposed to cascades, which describes multiple contagion events. We call this our *public opinion diffusion* (POD) model. This model allows media companies, who have played crucial roles in COVID misinformation [1, 5, 11], to be included in the study of opinion dynamics.

Through a simple cognitive function defined at the level of individual agents, our cognitive cascade model appears more expressive and ecologically valid than both cascade models that follow simple or complex contagion rules, and contagion models that do not simulate institution-driven cascades. Moreover, the motivations behind the DCC function demonstrate that simple and complex contagion rules cannot capture identity-related belief spread between individuals. Given findings from COVID misinformation studies, results from our POD model with DCC appear to align with population-level results reported by U.S. opinion polls —namely that beliefs about the virus remained starkly partisan [6–9], and hardly changed throughout 2020 [29]. This article is grounded in our experience with U.S. patterns of information, and the cases we cite are from the U.S. However, we note that misinformation has been on the rise worldwide, and in fact, may cross national boundaries as information is accessed on the Internet [71]. However, commonalities and differences and the extent that models for the US media ecosystem may generalize to different sociopolitical structures, or how models should be contextualized for different environments, will be left as a subject of future research. These preliminary results, which are in alignment with other similar emerging studies [72], hint at possible interventions and offer plenty of opportunities for future studies to be conducted using these methods.

## Background

### Social contagion

ABMs have been widely used to model *social contagion* effects—those which describe the process of ideas or beliefs spreading through a population [24]. Such models attempt to explain

possible processes underlying the spread of innovations [15, 73, 74], culture and ideology [27, 28, 72, 75–77], or unpopular norms [68]. These models are extensions of earlier work in sociology that theorized how social network structure and simple decisions, such as the threshold effect [47], may affect group-level behavior. Through more abundantly available computational power, these ideas can now be simulated and their implications can be analyzed.

There are two popular types of social contagion models used in ABMs: *simple*—also called independent cascade—and *complex contagion*—which has a proportional and absolute variation. Both model the spread of behaviors, norms, or ideas through a population. For simplicity, we will refer to behaviors or norms as "beliefs" going forward, as it is plausible to argue that both are generated by beliefs that an individual holds, explicit or otherwise.

**Simple contagion.** The simple contagion model assumes that behaviors or norms can spread in a manner akin to a disease [24, 37, 38, 44]. Simply being connected to an individual who holds a belief engenders a probability, $p$, that the belief may spread to you. This can even be true given different belief strengths or polarities for the same proposition. More formally, given two nodes in the model $u$ and $v$, with each having respective beliefs $b_u$ and $b_v$ at time $t$, when node $u$ is playing out its decision process (i.e. $u$ is the focal node and $v$ is the node exposing $u$ to a belief), the probability of adopting belief $b_v$ can be modeled as:

$$P(b_{u,(t+1)} = b_v \mid b_{u,t}) = p. \tag{1}$$

As opposed to many studies in innovation diffusion [15], we choose to argue in our formulation of contagion that every agent has a prior belief. Innovation diffusion studies often imagine any individual has no prior opinion about a new idea until they are "infected" with it. However, as we will illustrate below, we model beliefs on a spectrum, including belief in, against, and uncertainty about, a proposition. This departure from the epidemiological view of opinion diffusion allows us to argue for a prior, even if it is uncertainty.

It is important to note that in belief contagion models, the probabilities assigned to adopting a new belief given a prior one are over an event set of only two outcomes: adopt the new belief, or keep the prior one. Thus, if there are several possible beliefs to adopt from a set $B$, the sums of probabilities of adopting some $b_v$ given a prior of $b_u$, for all values in $B$, do not necessarily add to 1. Rather, because each agent interaction is only between two belief values, even if they both come from a larger set, the interaction is represented as a Bernoulli process. Each instance below in which we motivate probabilities of adopting a belief given a prior is adherent to the same logic.

Of course, since beliefs are not actually transmitted through airborne pathogens that incite infected individuals to believe something, there are abundant sociological hypotheses as to why this phenomenon may *appear* infectious [24]. There are other contagion models that have put forward alternative explanations.

**Complex contagion.** Complex contagion rather imagines that the spread of beliefs is predominantly governed by a ratio of consensus of those whom any agent is connected to [25, 42]. There are two major variations of complex contagion: what is typically called a *proportional threshold* contagion, and what we call an *absolute threshold* contagion. Proportional threshold contagion creates some $\alpha$ proportion of neighbors which must believe something for the ego agent $u$ to believe it. Absolute threshold contagion, on the other hand, may imagine some whole number $\eta$ of neighbors who must believe something in order for the ego $u$ to believe it [25, 47]. We choose to use the proportional threshold model for our examples and in-silico experiments. One of the most famous examples of this type of model, captured an a cellular automata paradigm, is in Schelling's segregation model [48]. Formally, given an ego $u$

and set of neighbors $N(u)$, the probability of adopting belief $b$ can be represented as:

$$P(b_{u,(t+1)} = b \mid b_{u,t}) = \begin{cases} 1, & \frac{1}{|N(u)|} \sum_{v \in N(u)} d(v,b) \geq \alpha, \\ 0, & otherwise \end{cases}, \tag{2}$$

where $d(v, b)$ is a simple indicator function which returns 1 if $b_v = b$—i.e. if neighbor $v$ believes $b$—and where $\alpha \in [0, 1]$ is a threshold indicating the ratio of believing neighbors necessary for $u$ to adopt the belief. In this model, the agent $u$ is guaranteed to adopt $b$ if a sufficient ratio of its neighbors believe $b$.

It may seem tempting to imagine, given the expected number of agents necessary to cross a repeated trial probability threshold from $p$, that the behavior of proportional threshold contagion can also be modeled with the "sufficient number of neighbors" idea. However, that notion does not capture the ratio effect that proportional threshold contagion models. Proportional threshold contagion says nothing about the *innate infectiousness* of any belief, but rather the infectiousness of *the connections surrounding* any agent.

The focus on a "portion" of believing neighbors being required to propagate a belief spawned new questions and investigations. This type of belief contagion has been argued to explain why some norms may spread despite them being disagreed with on an individual level, such as collegiate drinking behavior [68]. It elegantly captures phenomena associated with group dynamics such as peer pressure. It also can be used to model diffusion of health information or technological innovations [78].

**Cognitive contagion.** Both these contagion models can be generalized to allow heterogeneous sets of agents whose update rules are different for agents of different types (for example, more or less susceptible to infection). However, while these two popular types of contagion can effectively model some classes of belief contagion, others cannot be captured by their mechanisms—even with heterogeneous agents. Many simple and complex contagion models only imagine several states of belief, heavily influenced by epidemiological models: susceptible, infected, and removed. Models where there is an internal model of what a given individual agent *already believes* that dynamically affects what beliefs they spread and adopt cannot be described by either simple or proportional threshold contagion.

To address these problems, a class of what Zhang & Vorobeychik [15] call "cognitive agent models" exist. Based off of foundational work by Hegselmann & Krause [79], Deffuant et al. [80], DeGroot [43] and others, these models are often used to study group opinion dynamics when agents can influence each other in a more nuanced manner. Rather than simply allow agents to be "infected" by a belief or not, these models often place belief in any given proposition on a continuous spectrum. Agents then influence each other through opinion diffusion processes that can be tailored to a given cognitive or social phenomena. A generalization of the belief update process may be stated as:

$$P(b_{u,(t+1)} = b_v \mid b_{u,t}) = \beta(b_{u,t}, b_v), \tag{3}$$

$\beta$ can be a weighted update based on similarity of two agents' beliefs [27, 77], do nothing if the beliefs are too far away from each other [81], or be beholden to logical relations between beliefs [28]. Notably for our purposes, these models have been used to study polarization [27, 72, 75–77] and opinion dynamics given cognitive effects like dissonance reduction [81] or homophily [41].

## Cognitive cascade model

The goals of this work are twofold: (1) to apply empirically-grounded techniques from network and cognitive science to opinion diffusion models, and (2) to show that even the simplest resultant cognitive cascade model adds expressive power and leads to interesting dynamics of belief propagation that cannot arise in cascade models using the simple or proportional threshold contagion rules. To do so, we will lay out our cascade model and compare simulation results between using simple, proportional threshold, and cognitive contagion techniques.

### Agent contagion function

First, we can lay out which contagion models should be used for the agent-to-agent interactions occurring during a cascade. If our micro interactions are grounded in literature surrounding misinformation belief, we can analyze the ensuing macro effects knowing the model is grounded soundly. In our simple example, we consider belief in a single proposition $B$ (for example, $B$ could be, "COVID is a hoax," or, "mask-wearing does not help protect against spreading or contracting COVID"). In cognitive contagion models, as distinct from simple or proportional threshold contagion models, $u$'s probability of believing a message from $v$ is influenced by an internal model of $u$'s beliefs. For our simple cognitive cascade model, we model this internal prior initially as a single variable $b_u$, with $-1 \leq b_u \leq 1$. Without loss of generality, we use -1 to indicate strong disbelief and 1 to indicate strong belief.

Theoretically, $b_u$ can be a continuous variable with the interval from strong disbelief to strong belief, or it can take on discrete values. Inspired by frequently used 7-point scales to convey belief strength in public opinion surveys (e.g. [6, 11, 82], and justified by [83]), we choose 7 discrete, equally spaced values for belief in $B$ as follows: we represent the strength of the belief in proposition $B$ with elements from the set $B = \{b \mid 0 \leq b \leq 6\}, \ b \in \mathbb{Z}$; 0 represents strong disbelief, 1 disbelief, 2 slight disbelief, 3 uncertainty, 4 slight belief, 5 belief, and 6 strong belief. We note that our framework allows other resolutions of belief strength, from the discrete to approaching continuous, so we also explore these alternative model choices with additional in-silico experiments with lower and higher "resolutions" of belief: with $b$ able to take integer values between 0 and 2, 3, 5, 7, 9, 16, 32, and 64 to approach behavior over continuous beliefs. Results from those belief resolutions are shown in S9–S18 Figs. We find that the particular belief resolution of 7 was not exactly critical, and that nearby values (e.g. 5, 9, 13, and 16) produced very similar network dynamics. On the other hand, as $b$ approached a more continuous scale, we did not see exactly the same behavior, as some of our initial conditions changed in a way that made cascades more difficult. This implies that a realistic continuous model of beliefs might require different initial model parameters. More details can be found in S3 Text.

Importantly, this representation captures the polarity of the proposition as well: belief strength of the affirmative of $B$ (if $b \geq 4$), and the negation of $B$ (if $b \leq 2$). From here on, we will capture the idea of belief polarity—belief in or against a proposition $B$—by simply saying "belief strength."

We include this cognitive model for an individual agent within a message-passing ABM: At each time step $t$, agents have the chance to receive messages, to believe them, and share them with neighbors. We will further clarify the role of messages in spreading beliefs below, when we describe our diffusion model. But it should be noted upfront that regardless of being passed by a message, or by simple network connection exposure, we can compare beliefs from two agents, $u$ and $v$ the same way. We further note that cognitive science shows evidence that, for beliefs that are core to an individual's identity (such as political or ideological beliefs), exposure to evidence that is too incongruous with an individual's existing belief can cause individuals to

disregard evidence in an attempt to reduce cognitive dissonance [20]. Therefore, we later choose an update rule where an agent $u$ is only likely to believe messages when encoded belief values for proposition $B$ are not too far from $u$'s prior beliefs.

As a simple example, this could be represented by a binary threshold function. Given an agent $u$ with belief strength in $B$, $b_u$, and an incoming belief from $v$ with strength $b_v$, the following equation could govern whether agent $u$ updates its belief:

$$P(b_u = b_v \mid b_u) = \begin{cases} 1, & |b_u - b_v| \leq \gamma, \\ 0, & otherwise \end{cases}, \qquad (4)$$

where $\gamma$ is a distance threshold. Each agent has some existing belief strength in the proposition $B$, but will be unwilling to change their belief strength if a neighbor's belief strength is too far from theirs. There are similar functions motivated in contagion models centering dissonance [77, 81], and some which weight positive or negative influence differently [76]. We chose to weight positive or negative influence equally in this example, and subsequent contagion functions, to simplify the model and make its results more easily analyzable. Perhaps an agent $u$ who strongly believes the proposition ($b_u = 6$) will not switch immediately to strongly disbelieving it without passing through an intermediary step of uncertainty. Given a neighbor $v$ sharing belief $b_v = 0$, agent $u$ should not adopt this belief strength, because the difference in belief strengths is clearly greater than $\gamma$. Simple contagion would fall short because agent $u$ may simply randomly become "infected" with belief strength 0 by $v$ with some probability $p$. A proportional threshold contagion would similarly falter if agent $u$ were entirely surrounded by alters with belief strength 0. It would inevitably switch belief strengths regardless of some threshold $\alpha$ as in Eq (2).

As mentioned above, this manner of belief update could model the update of beliefs that are core to an individual's identity, such as political or ideological beliefs [31, 84, 85]. Rather than updating based on evidence presented, exposure to evidence that is too incongruous with an individual's existing belief may have no effect due to rationalization processes activated by cognitive dissonance [20].

In addition to the effects related to cognitive dissonance, there are other effects that have been reported to be involved in belief update. Pertinent to the misinformation literature, two that center the incoming belief itself are the illusory truth effect [23, 70] and the mere-exposure effect [86]. These effects emphasize that the number of exposures to a piece of information can motivate belief in it.

Regardless of effect, it is clear that some social contagion processes cannot be captured without modeling some sort of representation of an agent's cognition. For these reasons, we will extend work done in cognitive contagion models—particularly those modeling dissonance [81] and selective exposure [27, 72]—to capture the above effects. In general, given an agent $u$ with belief $b_u$, during its update step, the likelihood of updating its belief strength from $b_u$ to $b$, given their prior belief, can be captured by the cognitive contagion model in Eq 3. We graphically illustrate this process in Fig 1.

Because this equation is so general, there is a need to motivate a meaningful choice of $\beta$ function, and analyze how its effects differ from simple and proportional threshold contagion. There are obviously many choices for such a function, but the key lies in the fact that it compares beliefs between two agents, rather than being driven by network structure or mere chance. Below, we will describe our process of choosing a $\beta$ function in order to adequately model the misinformation effects that motivated our study.

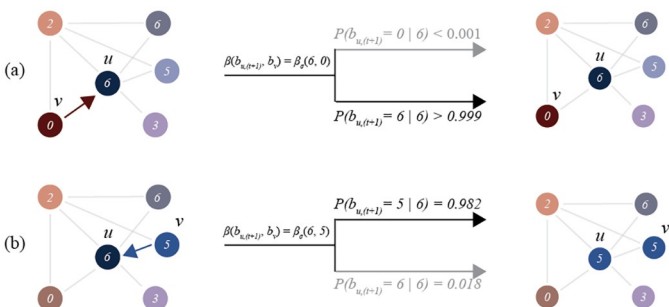

**Fig 1. A graphical illustration of cognitive contagion.** An illustration of cognitive contagion with the DCC contagion function described in Eq 6. (a) (Top) Given an agent $u$ with $b_u = 6$ and $v$ with $b_v = 0$, the chance of contagion is $< 0.001$. (b) (Bottom) Given an agent $u$ with $b_u = 6$ and $v$ with $b_v = 5$, the chance of contagion is 0.982.

## Institutional cascade model

Of course, a cognitive contagion model could be implemented on top of many different ABMs, so we will describe one that captures the misinformation problem, and that we can use over multiple experiments to arrive at a cognitive cascade model suited to the problem. Our *public opinion diffusion* (POD) ABM will be designed to capture the effects of misinformation spread by media companies, since in the aforementioned COVID misinformation studies, media played a pivotal role in people's belief or disbelief in safety protocols [82, 87, 88]. Moreover, in media studies, some scholars utilize frameworks such as Giddens [89] "theory of structuration"—which views media providers as macrolevel "structures" and individuals as microlevel "users" [52]—in order to model the ontological duality of media ecosystems.

Often, ABMs attempt to model so-called "levels" of social systems by grouping individual agents into a set—as in the case of a corporation being some hierarchical relation of individuals. Epstein persuasively argues that this individualistic method of modeling is not accurate, as, for example, a media organization's behavior depends not only on behaviors of a hierarchy of individual employees, but also on social conditions, governmental bodies, and more [90]. The POD model addresses this by including institutional influence in the form of media agents. We argue that an institution is not just a collection of individual agents, but an entirely different ontological entity, with separate incentives and influences, that still captures elements of media scholars framework for media ecosystems. Though, as will be clear below, our media agents are highly simplified, and could be made more complex in future studies. This addition is what makes our model a cascading model as opposed to a pure contagion model. Because the spread of messages starts with an institution—analogous to a media company—our contagion closely resembles cascade models used for analyzing media ecosystems [2, 5, 59]. A visual description of the model is included in Fig 2.

As previously mentioned, we will be using a message-passing ABM: at each time step $t$, agents have the chance to receive messages $m$ from the set of all possible messages $\mathcal{M}$—whose spread begins with media agents, which we call *institutional agents*—and to believe and share them with neighbors. We chose a message-passing model as opposed to the simple diffusion models often found in simple and proportional threshold contagion models because it allows us to capture the notion that beliefs are spread by explicit communication rather than simply by being connected to an agent.

Our model consists of $N$ agents in a graph, $G = (V, E)$ where each agent's initial belief strength $b_u$, $0 \leq u \leq N$ is drawn from a uniform distribution over the set of possible belief

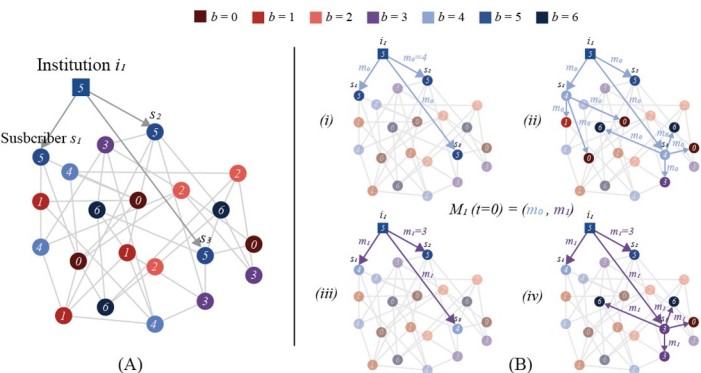

**Fig 2. A graphical illustration of one time step of the POD model.** In the left panel, (A) depicts the initial setup of a small network with institutional agent $i_1$ with subscribers $s_1, s_2, s_3$. All agents in the network are labeled with their belief strength. The right panel, (B) depicts one time step $t = 0$ of agent $i_1$ sending messages $M_1(t = 0) = (m_0, m_1)$. (i) shows the initial sending of $m_0 = 4$ to subscribers, and (ii) shows $s_1$ and $s_3$ believing the message and propagating it to their neighbors. (iii) and (iv) show the same for $m_1 = 3$, but only $s_3$ believes $m_1$.

strengths for proposition $B$, $B = \{b, 0 \leq b \leq 6\}$, $b \in \mathbb{Z}$. There is a separate set of institutional agents $I$—entirely different entities in the ontology of our model—which have directed edges to a set of "subscribers" $S \subseteq V$ if some parameter $\epsilon \leq |b_u - b_i|$, $u \in V, i \in I$—i.e. an agent in the network will subscribe to an institution if its belief strength from $B$ is sufficiently close to the belief strength of that institution. For all of our experiments, we will fix $\epsilon$ at 0. Institutional agents are designed to model media companies or notable public figures which begin the mass spread of ideas through the population. The belief strength of an institutional agent can be thought of as a perceived ideological "leaning" that would cause people with different prior beliefs to trust different media organizations.

At each time step, $t$, each institution $i$ will send a list of messages to each of its subscribers, represented by the function $M_i : t \to (m_0, m_1, \ldots, m_j)$, $m_j \in \mathcal{M}$, $j \geq 0$. In this simple example, the set of possible messages $\mathcal{M}$ will only encode one proposition, $B$, so for simplicity, we can set $\mathcal{M} = B$. Additionally, institutions will only send one message per time step. Whenever a message is received, an agent will "believe" it based on the contagion method being utilized, where $b_{mj}$ is the strength of belief in proposition $B$ encoded by the message, and $b_u$ is the agent's belief strength for $B$. If agent $u$ believes message $m_j$, then its belief strength is updated to be $b_{mj}$. When an agent believes a message, it shares the original message, $m_j$, with all its neighbors. It should be noted that because agent $u$ would change its belief strength to $b_{mj}$, agents will always share beliefs that are congruous to prior belief strengths—cohering to our cognitive contagion model outlined above. After a neighbor receives a message, the cycle continues: It has a chance to believe the message, and if believed, spread it to its neighbors. To avoid infinite sharing, each agent will only believe and share a given message once based on a unique identifier assigned to it when it is broadcast by the institutional agent.

Each time a message makes its way through the population, via some contagion method, we are capturing one cascade. By combining this cascading behavior with belief modeling typical of cognitive contagion models, we can synthesize the advantages of both disciplines for a more expressive and grounded model.

Our model and experiments were implemented using NetLogo [91] and Python 3.5. Code is made available on GitHub (https://github.com/RickNabb/cognitive-contagion).

## In-silico contagion experiments

### Experiment design

We wish to show that our cognitive cascade model can capture the observed effects of identity-based belief spread better than existing models of simple or proportional threshold contagion. To do so, we will lay out a series of in-silico (computer simulated) experiments to test each contagion method given the same initial conditions.

For each contagion method, we test three conditions where one institutional agent, $i_1$, attempts to spread different combinations of messages over time. We will refer to the first message set as *single*, as the institutional agent simply broadcasts one message for the entirety of the simulation; $M_i(t) = (6)$, $1 \leq t \leq 100$. The second set, we will call *split*, as the institution switches from messages of $M_i(t) = (6)$, $1 \leq t \leq 50$ to $M_i(t) = (0)$, $51 \leq t \leq 100$ halfway through the simulation. We call the final set *gradual* because the institution starts out spreading messages of $M_i(t) = (6)$, but at every interval of ten time steps, switches to $M_i(t) = (5)$, $M_i(t) = (4)$, etc. until finishing the last 30 time steps by broadcasting $M_i(t) = (0)$. We display these visually in Fig 3.

These specific sets of messages were chosen to expose distinct effects given each set, specifically with agent-to-agent cognitive contagion in mind. Based on research about how identity-related beliefs update, a proper cognitive contagion function would not allow agents with a belief strength significantly different from the message to update their belief. With the *single* message set, we wish to provide the simplest case: only one belief strength message is being spread. We predict that simple contagion will simply spread the messages to all agents, regardless of prior belief strength, and all agents will eventually update their belief strength to that in the message. We also anticipate that proportional threshold contagion, being so reliant on prior belief strengths of an agent's neighbors, will not be so straightforward. Assuming a nearly uniform distribution of agent neighbor belief strengths, the threshold chosen would likely make the difference between all agents updating their beliefs, or no agents updating their beliefs. A proper cognitive contagion function that captures our desired effect should see only belief updates from agents whose belief strengths are already close to the belief strength encoded in the message.

The *split* message set, on the other hand, should have different effects. We predict that simple contagion will see all agents believe the first belief, then all agents believe the second, while proportional threshold contagion may spread the initial belief but not the second. For cognitive contagion, if the function we choose successfully models our target phenomena, only agents within the same threshold as in the *single* condition should believe the first message,

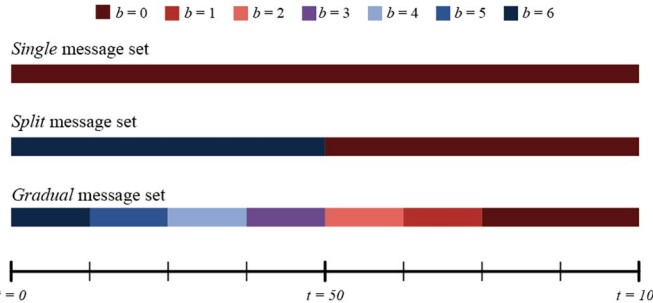

**Fig 3. A graphical illustration of the different message sets.** A visual depiction of the different message set conditions used in our in-silico experiments: *single* (top), *split* (middle), and *gradual* (bottom), set against a 100-step simulation from $t = 0$ to $t = 100$.

then virtually no agents should believe the second—except a few who may switch based on exposure effects.

Finally, we predict that the *gradual* message set should be the only which allows cognitive contagion to sway the entire network. Because agents will only believe messages that are relatively close to their prior beliefs, it logically follows that the only way to move beliefs from one pole to another is incrementally. We further anticipate that simple contagion will sway the entire agent population to adopt the belief strength of each message in turn, and that proportional threshold contagion may sway the entire population to one specific belief strength, but then not be able to change any agent belief strengths after such a contagion.

We also keep certain contagion variables static between experiments and conditions. In each case, we will fix the simple contagion probability, $p$ to be 0.15, and fix $\alpha$, the proportional threshold contagion neighbor threshold to be 0.35. Parameters are fully laid out in Table 1. The former was chosen to allow a slower spread of belief strengths, as higher values would make the spread too fast to properly analyze. The latter was chosen as it, too, would ideally avoid being so low that contagion happens immediately and in all cases, or so high that it never occurs. In preliminary experiments, the chosen values best satisfied these goals. We further explain the process we underwent to choose these values in S1 Text and S1 Table.

As a final experimental condition to vary independently of message set, we will run experiments on a host of different graph topologies, keeping the number of nodes and prior distribution of belief strengths constant. We test each contagion method on four networks topologies:

- The Erdős-Rényi (ER) random graph [92]

- The Watts-Strogatz (WS) small world network [93]

- The Barabási-Albert (BA) preferential attachment network [94]

- The Multiplicative Attribute Graph (MAG) [95]

We explicate rationale behind choosing each below. Running experiments across different graph topologies should allow us to determine how much graph structure affects cascades. We anticipate that cascades using simple contagion may not vary much over graph type, that those with proportional threshold contagion will vary the most because it is the most dependent on neighborhood structure, and that cognitive contagion should vary least, as it relies more on single instances of neighbor belief strengths rather than an aggregate. Additionally, because

**Table 1. Parameter values for in-silico contagion experiments.**

| Parameter | Value | Description |
|---|---|---|
| *Contagion Methods* | | |
| $p$ | 0.15 | The chance of spread for simple contagion. |
| $\alpha$ | 0.35 | The ratio for proportional threshold contagion. |
| $\beta(b_{u,(t+1)}, b_v)$ | $\frac{1}{1+e^{\alpha(\|b_{u,t}-b_v\|-\gamma)}}$ | The cognitive contagion function (DCC, Eq (6)) we use. |
| $\alpha, \gamma$ | $\alpha = 4, \gamma = 2$ | Scale and translation values for the DCC function, $\frac{1}{1+e^{\alpha(\|b_{u,t}-b_v\|-\gamma)}}$ |
| *POD Model* | | |
| $N$ | 500 | The number of networked agents. |
| $\|I\|$ | 1 | The number of institutional agents. |
| $\epsilon$ | 0 | The maximum distance between agent and institution beliefs necessary to be a subscriber. |
| $T$ | $\{t, 0 \leq t \leq 100\}$ | The set of timesteps. |

the model has stochastic elements in the initial distribution of agent beliefs, at each potential contagion step governed by the contagion function, and in the randomness of the networks constructed, experiments are run from ten to one hundred times and results are shown as averages over the total simulations, with variances displayed in supplemental materials S1 and S2 Figs where applicable. We display results aggregated over 10 simulations here, as there was not significant variation between results from 10, 50, or 100 simulations, and include results from 50 and 100 simulations with justification for using 10 in S3 Text, S2 Table, and S23 Fig.

But first, we need to choose a cognitive contagion function that best suits our empirically-based goals. After choosing such a function, it will be compared against simple and proportional threshold contagion in each condition.

## Motivating choice of $\beta$ functions

Unsurprisingly, depending on the choice of $\beta$, the network should display significantly different dynamics. That choice should depend on what type of phenomena is being modeled. We will explore a variety of choices for this function and compare the outcome of their respective cognitive contagion against the qualitative target phenomena. Other works studying cognitive contagion have motivated similar dissonance-related contagion functions [76, 81], treating dissonance as a weighted update based on belief distance.

We want to follow suit, and tune the choice of function to capture the effect of updating beliefs core to an individual's identity: in a manner where incoming belief strengths must be close to existing belief strengths to yield an update. If possible, it would also be useful to be able to choose a function which also captures aspects of exposure effects: that beliefs do have a small chance to update that grows as the agent is exposed to a belief over time. However, agents should prioritize the dissonance effect. That is, they should only experience an exposure effect if incoming messages are already somewhat close to existing beliefs. The dissonance effect should take precedence. Our function will differ, however, in our discrete belief updating— moving from one discrete value to another rather than meeting in between as in prior work.

Keeping these two target phenomena in mind, we explore three classes of functions: linear, threshold, and logistic.

**Linear functions.** To begin with perhaps the simplest function, an inverse linear function would capture the effect of making beliefs that are further apart have a smaller probability of updating. Moreover, we can generalize this inverse function by adding some parameters to add a bias and scalar to the denominator. The equation comes out as follows:

$$P(b_{u,(t+1)} = b_v \mid b_{u,t}) = \beta(b_{u,t}, b_v) = \frac{1}{\gamma + \alpha|b_{u,t} - b_v|} \tag{5}$$

In this equation, $\gamma$ becomes a parameter to add bias towards being reluctant to update beliefs, and $\alpha$ similarly decreases the probability of update as it increases. This turns out to be a useful formulation, because if we set $\gamma$ and $\alpha$ to be very low, then the agent becomes relatively more "gullible." Conversely, setting $\gamma$ and $\alpha$ to be high would make the agent "stubborn." We expect that the "stubborn" agent will be most desirable for our purposes of modeling cognitive dissonance.

To compare parameterizations of the inverse linear function, we contrast, on an Erdős-Rényi random graph, $G(N, \rho) = (V, E)$ with $N = 250$, $\rho = 0.05$, in our POD model setup described above, a relatively "gullible" agent function ($\gamma = 1, \alpha = 0$) to a "normal" function ($\gamma = 1, \alpha = 1$), and to a "stubborn" ($\gamma = 10, \alpha = 20$) function. We additionally display results for only the *split* message set, though results for others can be found in S3–S8 Figs. Results are displayed in Fig 4.

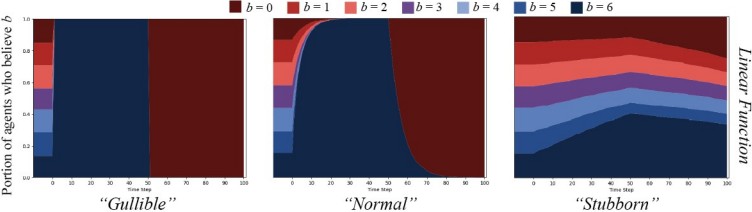

**Fig 4. Linear cognitive contagion function result comparisons.** The *split* message set on an ER random graph, $N = 250$, $\rho = 0.05$, for agents updating their beliefs based on the inverse linear cognitive contagion function in Eq (5). Graphs display percent of agents who believe $B$ with strength $b$ over time. The left graph shows agents parameterized to be "gullible" ($\gamma = 1$, $\alpha = 0$); the middle shows "normal" agents ($\gamma = 1$, $\alpha = 1$), and the right, "stubborn" agents ($\gamma = 10$, $\alpha = 20$).

As expected, the results show that the "gullible" agents simply believe everything. The "normal" agents take a bit longer to all update their belief strengths to that of the messages broadcast, but eventually do. Importantly, when all agents' belief strength for $B$ is $b = 6$ after the first 50 time steps, they all quickly switch over to $b = 0$, which does not fit the cognitive dissonance effect we are trying to model. The "stubborn" agent case is the closest to what we are seeking. Only the agents who are already closest to $b = 6$ believe the first message over time, with some effect on agents with more distant beliefs. After the messages switch polarity halfway through, the agents whose strength of belief is $b = 6$ do drop significantly, but less so than in the other conditions. This effect seems closer to the dissonance mixed with exposure effects that we desire: beliefs are less likely to change as they are far away, but there is a chance to change with many messages over time.

**Threshold functions.** Next, we evaluate the behavior of threshold functions and compare to our desired effect. Threshold update functions are used in opinion diffusion ABMs, most notably in the HK bounded confidence model [79, 81, 96]. We anticipate that these functions will capture the desired dissonance effect. The function can be parameterized as we already motivated in Eq (4), with $\gamma$ serving as the threshold.

Using the same formulations as above, with "gullible" ($\gamma = 6$), "normal" ($\gamma = 3$), and "stubborn" ($\gamma = 1$) agents, we find results on the same graph structure as displayed in Fig 5.

These results confirm our anticipations, and perfectly capture the effect of only updating if incoming messages are within a certain distance of existing beliefs. However, the threshold function leaves no possibility of update to capture the repetition effects of mere exposure or illusory truth. The probability of updating is either 0 or 1, which loses much of the nuance of the actual phenomena.

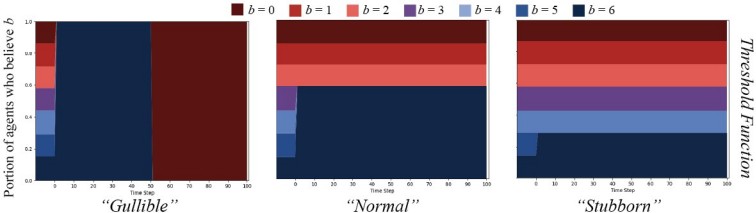

**Fig 5. Threshold cognitive contagion function result comparisons.** The *split* message set on an ER random graph, $N = 250$, $\rho = 0.05$, for agents updating their beliefs based on the threshold cognitive contagion function in Eq (4). Graphs display percent of agents who believe $B$ with strength $b$ over time. The left graph shows agents parameterized to be "gullible" ($\gamma = 6$); the middle shows "normal" agents ($\gamma = 3$); and the right, "stubborn" agents ($\gamma = 1$).

**Sigmoid functions.** Finally, we will test a logistic function—specifically a sigmoid function, as we are attempting to capture probabilities and the range of the sigmoid is [0, 1]. Sigmoid functions are commonly used in neural network models because they capture "activation" effects arguably akin to action-potentials in biological neurons, and rein in outputs so they do not explode while learning [97]. This property is useful for our purposes as well. We formulate our sigmoid cognitive contagion function as follows:

$$\beta(b_{u,(t+1)}, b_v) = \frac{1}{1 + e^{\alpha(|b_{u,t} - b_v| - \gamma)}} \tag{6}$$

In this equation, $\alpha$ and $\gamma$ control the strictness and threshold, respectively. As $\alpha$ increases, the function looks more like a binary threshold function, and restricts any significant probability to center around $\gamma - 1$. $\gamma$ controls the threshold value by translating the function on the $x$ axis. Though, in a sigmoid function, the value at $x = \gamma$ will always be 0.5, so if one wishes to guarantee belief update given a threshold $\tau$, it must be set as $\tau = \gamma + \epsilon$ where $\epsilon \propto \frac{1}{\alpha}$. We use this strategy to set our $\gamma$ values throughout our in-silico experiments.

Given the same experiments as above, the sigmoid function parameterized for different agent types ("gullible" ($\alpha = 1$, $\gamma = 7$), "normal" ($\alpha = 2$, $\gamma = 3$), and "stubborn" ($\alpha = 4$, $\gamma = 2$)) yields results as show in Fig 6.

These results seem to display characteristics of both the inverse linear function and the threshold function. The "gullible" agents, as always, believe everything but with a softer transition than for the linear or threshold functions. "Normal" agents are the first indication that we are getting closer to our desired effect. After an initial widespread uptake of belief strength $b = 6$ in the first half of the simulation, some agents begin to believe $b = 0$, but the population that decreases the most to engender that gain seem to be agents with belief strength $b = 1$. Though, from our model, some agents with strength $b = 6$ must have believed $b = 0$ messages, because if they did not, the messages would never have been shared and made it to $b = 1$ agents—the institutional agents are only connected to $b = 6$ agents as $\epsilon = 0$.

Finally, the "stubborn" agents seem to best capture our desired effect. Initially, $b = 6$, $b = 5$, and $b = 4$ agents are the only who update beliefs. There is also a small effect where some $b = 3$ agents update to $b = 6$, capturing the exposure effects combined with a dissonance effect. Importantly, when messages switch to $b = 0$, none of the $b = 6$ agents update their beliefs. The exposure effects would not work as the dissonance effect would take primacy.

These agents act in a way akin to what we observe from cognitive literature [20, 23], albeit in a highly simplified manner: an agent who "strongly disbelieves" in something like COVID mask-wearing will likely only be swayed by a message that "disbelieves" or is "uncertain" about the belief. On the individual level, a maximum two relative magnitudes of belief separation, with decreasing probabilities as distance increases, seems to qualitatively match empirical

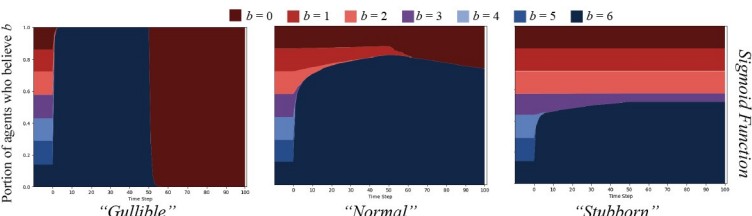

**Fig 6. Sigmoid cognitive contagion function result comparisons.** The *split* message set on an ER random graph, $N = 250$, $\rho = 0.05$, for agents updating their beliefs based on the sigmoid cognitive contagion function in Eq (6). Graphs display percent of agents who believe $B$ with strength $b$ over time. The left graph shows agents parameterized to be "gullible" ($\alpha = 1$, $\gamma = 7$), the middle shows "normal" agents ($\alpha = 2$, $\gamma = 3$), and the right, "stubborn" agents ($\alpha = 4$, $\gamma = 2$).

work. In our simulations using more than 7 points on a belief spectrum, this argument can still be held by setting the equivalent belief "markers" along the spectrum, and using those to scale the contagion function.

Given these initial experiments, it seems most reasonable to choose the "stubborn" sigmoid cognitive contagion function as that which best captures our desired effects. We will use this *defensive cognitive contagion* (DCC) function in the rest of our experiments as we compare the effects of cascades with cognitive contagion to those with simple and proportional threshold contagion.

## Comparing contagion methods

Now that we have selected a cognitive contagion function that best captures the effects we wish to model, it is necessary to compare cascade results of this function to those from simple and proportional threshold contagion in our cascading POD model. We will investigate the way that these different contagion methods manifest effects on different network structures. For simplicity, we will henceforth refer to each cascade-contagion function combination as *simple cascades*, *proportional threshold cascades*, and *cognitive cascades*, respectively. Parameter values used in these comparisons are listed in Table 1. In addition to significant effects based on the choice of the $\beta$ function, we expect that effects will also significantly differ based on the structure of the network. The structure will determine which ideas reach agents, and which do not, and thus should affect the final outcome of belief distribution over the network.

We will test each cascade method on five types of networks: the Erdős-Rényi (ER) random graph [92], the Watts-Strogatz (WS) small world network [93], the Barabási-Albert (BA) preferential attachment network [94], and the Multiplicative Attribute Graph (MAG) [95]. Each network has distinct properties that will affect how the cascading contagions play out. Additionally, we will test each message set for each network type to explore the effects of different influence strategies.

First, we will qualitatively analyze the aggregate effects on the diffusion of beliefs, and subsequent belief strength updates, across the entire network. After doing so for each network topology, we will analyze the cascading behavior itself.

### Cascades on ER random networks

We will begin with cascades on Erdős-Rényi [92] random networks $G(N, \rho) = (V, E)$ where $\rho = 0.05$ and $N = 500$. Note that $\rho$ is not the chance of simple contagion, $p$, but the chance that two agents connect in the random graph. This graph type was chosen as a baseline to compare others to, as is standard in network science. Results of the simple and proportional threshold cascades are shown in S19 Fig.

Results from the simple cascade experiments show that in each message set condition, the belief strength being spread pervaded the entire network in all cases. Moreover, the strength of beliefs broadcast were adopted by the population very quickly. In the case of proportional threshold cascades, the initial distributions of belief strengths did not change in any message set condition.

Cognitive cascades on the ER graphs yielded markedly different results. As depicted in Fig 7, both the *single* and *split* message sets were only able to sway agents that started with $b = 6$, $b = 5$, or $b = 4$, with what appears to be a few $b = 3$ agents persuaded. Importantly, no agents were swayed after the messaging change in the *split* condition. The *gradual* message set is the only that was able to sway all agents over to $b = 0$.

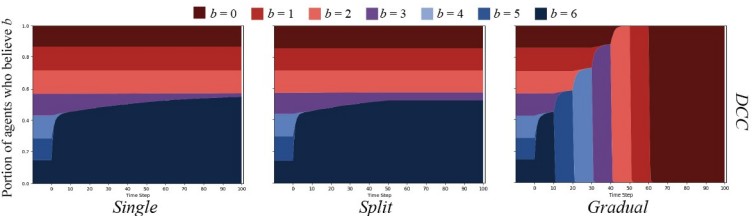

**Fig 7. DCC results on ER random networks.** DCC on ER random networks with $N = 500$, and connection chance $\rho = 0.05$. Graphs show the percent of agents who believe $B$ with strength $b$ over time.

## Cascades on WS small world networks

Our second set of experiments were conducted on Watts-Strogatz [93] small world networks, $G(N, k, \rho) = (V, E)$, where $N = 500$, $k = 5$, and $\rho = 0.5$. In this formulation, $k$ is the number of initial neighbors any node is connected to, and $\rho$ is the chance of rewiring any edge. We chose this graph topology because small world networks exhibit some attributes of real-world social networks (low diameter and triadic closure). Results of simple and proportional threshold cascades on these WS graphs are shown in S20 Fig.

Simple cascade results showed largely the same pattern as in the ER random graph, but a significantly slower spread through the population. Interestingly, proportional threshold cascades were successful on the WS graphs, but with a high amount of variance over simulations (variance shown in S1 Fig).

Results from cognitive cascade experiments closely match those from the ER random graph experiments. These are displayed in Fig 8. The population displayed the same patterns given a significantly different graph structure, which begs explanation. We will discuss this further below.

## Cascades on BA preferential attachment networks

We continued our experiments by testing Barabási-Albert [94] preferential attachment networks, $G(N, m) = (V, E)$, where $N = 500$ and $m = 3$, where $m$ represents the number of edges added with each newly added node. This network type was also chosen because of its properties that closely resemble real-world social networks (low diameter, power law degree distribution). Results are shown in S21 Fig.

Again, simple cascade results are similar to those of the WS graphs: slower spread than in ER random graphs, but faster than in WS graphs. In this case, spread is facilitated by the power law distribution of node degree—even a few nodes with high degree believing the message can have an outsize effect in spreading quickly to the outskirts of the network [98]. Also

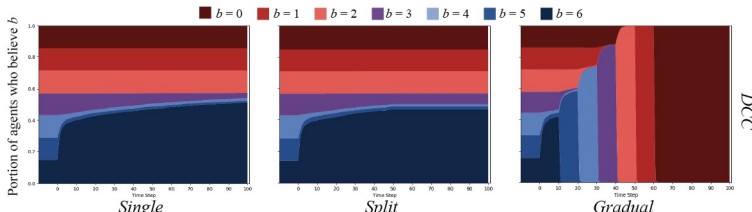

**Fig 8. DCC results on WS small world networks.** DCC on Watts-Strogatz small world networks with $N = 500$, initial neighbors $k = 5$, and rewiring chance $\rho = 0.5$. Graphs show the percent of agents who believe $B$ with strength $b$ over time.

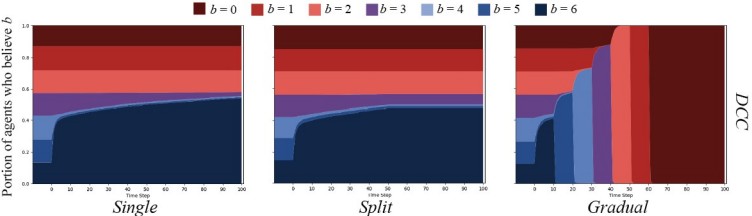

**Fig 9. DCC results on BA preferential attachment networks.** DCC on a Barabási-Albert preferential attachment network with $N = 500$, and added edges $m = 3$. Graphs show the percent of agents who believe $B$ with strength $b$ over time.

interestingly, proportional threshold cascades appear to have slight effects on these graphs, with the *gradual* message set having most effect. Proportional threshold cascade results also showed significant variance (shown in S2 Fig).

Results from cognitive cascade experiments were again similar to those of the ER random and WS graphs—most similar to results from the WS graph. Results are shown in Fig 9.

## Cascades on MAG networks

Finally, we tested cascades on the Multiplicative Attribute Graph [95] with an affinity matrix $\Theta_b$ that yielded a graph with very high homophily. We chose this graph topology because real world social networks are homophilic—people who have similar interests tend to connect. Testing a highly homophilic graph (higher than in a real social network) can allow us to test the extreme case of communities in silos based on their belief strength. The affinity matrix was constructed as follows:

$$\Theta_b = (\theta_{ij}) \in \mathbb{R}^{m \times n} = \frac{1}{1 + 50(j - i)^2} \tag{7}$$

$$= \begin{bmatrix} 0.167 & 0.018 & 0.005 & 0.002 & 0.001 & 0.0008 & 0.0006 \\ 0.018 & 0.167 & 0.018 & 0.005 & 0.002 & 0.001 & 0.0008 \\ 0.005 & 0.018 & 0.167 & 0.018 & 0.005 & 0.002 & 0.001 \\ 0.002 & 0.005 & 0.018 & 0.167 & 0.018 & 0.005 & 0.002 \\ 0.001 & 0.002 & 0.005 & 0.018 & 0.167 & 0.018 & 0.005 \\ 0.0008 & 0.001 & 0.002 & 0.005 & 0.018 & 0.167 & 0.018 \\ 0.0006 & 0.0008 & 0.001 & 0.002 & 0.005 & 0.018 & 0.167 \end{bmatrix} \tag{8}$$

To measure homophily, we used a simple measure of the global average neighbor distance given the $b$ value of each node, and compared against a random ER graph. The measure is detailed in Eq (9):

$$h(G = (V, E)) = \frac{\sum\limits_{v \in V} \sum\limits_{u \in N(v)} |b_u - b_v|}{2|V|^2}, \tag{9}$$

where $N(v)$ is a function that returns neighbors $u \in V$ of $v$. Over ten ER random graphs with $N = 500$ and $\rho = 0.05$, the mean average neighbor distance was 2.30 with a mean variance of 0.349. Over ten MAG graphs generated with $\Theta_b$, the mean average neighbor distance was 0.31

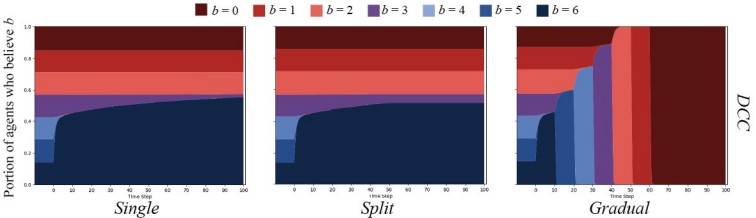

**Fig 10. DCC results on homophilic MAG networks.** DCC on homophilic MAG networks with $N = 500$, and $\Theta_b$ detailed in Eq (8). Graphs show the percent of agents who believe $B$ with strength $b$ over time.

with a mean variance of 0.02. Results from simple and proportional threshold cascades on these homophilic MAG graphs are shown in S22 Fig.

As it turns out, a high degree of homophily did not appear to make a significant difference to any cascade patterns. The patterns generated from simple and proportional threshold cascades appear similar to those from the ER random graphs. The same is true for cognitive cascade results, which are depicted in Fig 10.

## Analysis of results

### Belief change

To quantify the relationships we observe qualitatively between the cascading contagion patterns, we can examine correlations between results. We choose to illustrate these patterns using only data from the *single* messaging pattern, as differences in this pattern appear to be the smallest within cascade methods. By arguing from the standpoint of the apparent most similar results, it implies that results that are more different given simple and proportional threshold cascade methods—while DCC results are still quite similar—further prove the stability of the cascade method.

Table 2 displays results within cascade methods (simple, proportional threshold, and DCC) between pairs of graph types. We measure various correlations between belief results from these pairs. Details of the methods by which we performed the tests can be found in S4 Text. Briefly, $\bar{r}$ denotes an average Pearson coefficient across belief values; and $\bar{\chi}^2$ denotes a version of the $\chi^2$ test applied to time series data, testing for independence between the two

**Table 2. Correlation test results measuring similarity between cascades on graph type pairs, by cascade method.**

| Measure | ER-WS | ER-BA | ER-MAG | WS-BA | WS-MAG | BA-MAG |
|---|---|---|---|---|---|---|
| *Simple* | | | | | | |
| $\bar{r}$ | 0.751 | 0.869 | **0.999** | 0.973 | 0.750 | 0.868 |
| $\bar{\chi}^2$ | 0.010 | 0.208 | 1.000 | 0.040 | 0.010 | 0.208 |
| *Proportional Threshold* | | | | | | |
| $\bar{r}$ | 0.529 | 0.649 | $\emptyset$ | 0.899 | $\emptyset$ | $\emptyset$ |
| $\bar{\chi}^2$ | 0.020 | 0.040 | 1.000 | 0.020 | 0.020 | 0.040 |
| *DCC* | | | | | | |
| $\bar{r}$ | **0.964** | **0.983** | 0.981 | **0.989** | **0.970** | **0.917** |
| $\bar{\chi}^2$ | **0.327** | **1.000** | 1.000 | **1.000** | **0.327** | **1.000** |

Correlation tests within cascade method, between graph type pairs. Bolded values are the strongest correlations across cascade methods for a given test. $\emptyset$ denotes that a test could not be performed because no changes occurred from initial conditions on one or both graphs.

graphs' belief distributions at each time step. For all test values, a higher value indicates stronger correlation.

It is clear from correlation test results that the DCC method yields the strongest correlations between graph topologies. This supports what looks apparent qualitatively from graph results. While for simple and proportional threshold cascade, correlations are weak or hardly present, almost all values for DCC are consistently strong. Intuitively, we may expect that graph structure would affect any cascading contagion method, as is the case for simple and proportional threshold. However, this analysis reveals that when using the DCC function, effects of structure are greatly diminished in most cases.

## Cascades

Results across different graph topologies further support our motivations for introducing cognitive cascade models. These results are encouraging because the entire motivation for our model takes out the structural dependencies of proportional threshold cascade, and replaces simple cascades' random chance of spread with one motivated by what agents already believe. Thus the key factors determining the spread of messages and change in beliefs in our cognitive cascade model are:

1. Whether or not any given agent is exposed to some message;

2. How many times an agent is exposed to similar messages; and

3. The difference between agent beliefs and that message.

These results qualitatively match what has been observed in misinformation literature. Even when exposed to factual or scientific evidence (e.g. that wearing masks would mitigate the COVID-19 pandemic), people who are already skeptical of mask-wearing are not able to be swayed. They often instead rationalize their existing beliefs [2, 22, 31, 85]. Additionally, mass-exposure to a given message still has a chance to sway agents in our model—proportionally to how distant that message is to agent beliefs. This captures the illusory truth [23, 70] and mere-exposure effects [86].

We can also quantify this result by analyzing the cascading behavior for any given message. For any message $m_j$ from a list $M_i : t \rightarrow (m_0, m_1, \ldots, m_j), m_j \in \mathcal{M}, j \geq 0$ sent by an institution $i$ at time step $t$, there will be a probability that any agent $u$ will believe the message, which depends on the factors listed above. In terms of our model,

(1) becomes the probability of $m_i$ being received and believed by any neighbor $N(u)$ of $u$;

To travel from institution $i$ to agent $u$, a message must follow a directed path through the graph, $w_{iu} = (v_1, v_2, \ldots, v_n)$ where $v_1 = i$ and $v_n = u$. The probability of a message being passed down the entire path can be expressed as:

$$\mathcal{P}(m_j, w) = \prod_{v \in w} \beta(v, m_j). \tag{10}$$

To then properly represent (1), we can limit $w_{iu}$ to end at neighbors of $u$.

The next step requires us to determine how many neighbors $N(u)$ of $u$ believe $m_j$—as they would then subsequently propagate the message to $u$. Therefore,

(2) becomes $|N_\beta(i, u, m_j)|$, the number of neighbors of $u$ who are likely to believe $m_j$ coming from $i$.

However, we do not know ahead of time which neighbors will actually believe $m_j$ as the model is stochastic. We can argue that a probability within some $\delta$ will suffice. We can

represent $N_\beta(i, u, m_j)$ as:

$$N_\beta(i, u, m_j) = \{ \ v \ | \ v \in N(u) \wedge \mathcal{P}(w_{iu}, m_j) \geq 1 - \delta \ \}, \tag{11}$$

Because in the POD model, agents can only believe and share $m_j$ once, to determine how many neighbors of $u$ would share the message, we must choose a set of non-overlapping paths $\mathcal{W}^*$ from $i$ to neighbors of $u$—moreover, one that maximizes total path probabilities:

$$\mathcal{W}_{iu}^* = \max_w \prod \mathcal{P}(m_j, w) \ \{ \ w_{iu} \ | \ \bigcup_{w_{iu}}(\cdot) = \emptyset \ \} \tag{12}$$

The algorithmic formulation of such a process would best be captured in future work.

Regardless of methods, it stands that $\mathcal{P}(m_j, w)$ is crucial in determining whether any agent $u$ will have a chance of receiving a message and believing it. This result can help explain why simple and proportional threshold cascades showed such variation across graph types, but cognitive cascades did not. Given the probabilities of adopting belief strengths $b_u$ given a prior belief of $b_v$ for DCC shown in Table 3, it becomes clear that if any $\beta(v, m_j)$ is given a belief strength difference of 3 or higher, then the entire chain's probability $\mathcal{P}(m_j, w)$ will collapse to very close to 0.

To satisfy (1), a path of agents with belief strengths at most distance 1 away from the message would reliably transmit it with high probability. If any agents in the path have a belief of distance 2 from the message, the transmission probability would decrease; halving the probability with each agent of distance 2. Compare this with simple cascades, where every agent has a flat 0.15 probability of sharing, and the path probability converges close to 0 in only two steps; or proportional threshold cascades where if any agent in the path does not meet the threshold of 0.35, the path probability immediately collapses to 0.

Taking these criteria for (1) into account, Table 3 also makes clear that to satisfy (2) and (3), the chain may only need to end with one agent—i.e. $|\mathcal{W}_{iu}^*| = 1$. If the message belief strength is distance 0 or 1 from $u$, then there is already a near guaranteed chance that $u$ will believe the message after receiving it only once. Conversely, in any quantitative analysis of which agents may believe a message, we can exclude with high confidence all agents with a belief strength difference of 3 or higher from consideration, as their chances of believing the message even if it made it to them would be near zero.

Therefore, to quantitatively demonstrate why the cognitive cascade results were so stable across random graph types, we can show, for a randomly selected agent $u$ with belief strength $b_u$, the percentage of 100 random generations of the graph which yield at least one path of agents $v$ entirely with $b_v$ distance at most $\tau$ away from a message with belief strength $b_{mj}$.

**Table 3. Probabilities of adopting beliefs $b_u$ given prior $b_v$ using DCC.**

| $b_u/b_v$ | 0 | 1 | 2 | 3 | 4 | 5 | 6 |
|---|---|---|---|---|---|---|---|
| 0 | 0.999 | 0.982 | 0.500 | 0.018 | <0.001 | <0.001 | <0.001 |
| 1 | 0.982 | 0.999 | 0.982 | 0.500 | 0.018 | <0.001 | <0.001 |
| 2 | 0.500 | 0.982 | 0.999 | 0.982 | 0.500 | 0.018 | <0.001 |
| 3 | 0.018 | 0.500 | 0.982 | 0.999 | 0.982 | 0.500 | 0.018 |
| 4 | <0.001 | 0.018 | 0.500 | 0.982 | 0.999 | 0.982 | 0.500 |
| 5 | <0.001 | <0.001 | 0.018 | 0.500 | 0.982 | 0.999 | 0.982 |
| 6 | <0.001 | <0.001 | <0.001 | 0.018 | 0.500 | 0.982 | 0.999 |

Probabilities given by $\beta(b_u, b_v)$ for the DCC function, described in Eq (3).

**Table 4. Probabilities of agents with belief $b_u$ having a strongly contagious DCC path to it from an institutional agent.**

| ($\tau = 1$) | $b_u = 0$ | $b_u = 1$ | $b_u = 2$ | $b_u = 3$ | $b_u = 4$ | $b_u = 5$ | $b_u = 6$ |
|---|---|---|---|---|---|---|---|
| ER | 0.91 | 0.9 | 0.97 | 0.95 | 1.0 | 1.0 | 1.0 |
| WS | 0.51 | 0.56 | 0.59 | 0.7 | 0.64 | 0.62 | 1.0 |
| BA | 0.73 | 0.66 | 0.66 | 0.75 | 0.63 | 0.75 | 1.0 |
| MAG | 0.13 | 0.2 | 0.37 | 0.54 | 0.8 | 1.0 | 1.0 |
| ($\tau = 2$) | $b_u = 0$ | $b_u = 1$ | $b_u = 2$ | $b_u = 3$ | $b_u = 4$ | $b_u = 5$ | $b_u = 6$ |
| ER | 0.98 | 0.99 | 0.97 | 1.0 | 1.0 | 1.0 | 1.0 |
| WS | 0.65 | 0.8 | 0.76 | 0.72 | 0.76 | 0.81 | 1.0 |
| BA | 0.84 | 0.89 | 0.84 | 0.82 | 0.89 | 0.88 | 1.0 |
| MAG | 0.29 | 0.29 | 0.45 | 0.84 | 0.99 | 1.0 | 1.0 |

Proportion of 100 random graphs ($N = 500$) with at least one path leading from the institutional agent $i$ to a randomly selected node $u$ with belief strength $b_u$, where each agent $v$ in the path has belief strength $|b_v - b_{mj}| \leq \tau$, and $b_{mj} = 6$.

Moreover, we can show this for all potential values of $b_u$, keeping the belief strength of the message constant, $b_{mj}$, at 6 (quantifying the *single* message condition). Importantly, this path does not include $u$ because for $b_u = 3$ and below, the distance of $b_u$ to $b_{mj}$ would always be too high; thus, our paths lead to neighbors of $u$. These paths were found by assigning edge weights equal to the distance between the message and the belief strength of the source node in the directed pair, and running Dijkstra's weighted shortest path algorithm with $i$ as the source and $u$ as the target. Results are displayed in Table 4.

When $\tau$ is 1 the path yields an almost guaranteed probability of the message reaching $u$. When $\tau$ equals 2, the path can yield a range of chances to reach $u$—depending on how many distances of 2 there are in the path, each inserting a probability of 0.5 into the total product. However, compared to the path probability for cascades using simple or proportional threshold contagion—where the former is a path entirely of probabilities of 0.15, and the latter requiring meeting a threshold of 0.35 for each agent in the path to even have a non-zero chance—both path types under DCC yield significantly higher probabilities of messages reaching target agents.

Moreover, the analysis shows that across graph topologies, where there are high proportions of both path types present, there is a high likelihood for messages to reach agents of *all* belief strengths. Particularly for Erdős-Rényi random graphs, both path types are almost always present. Both Watts-Strogatz and Barabási-Albert networks show lower, but still high proportions of both path types being present. This likely accounts for the slight variations in cognitive cascade results displayed in graphs above. Predictably, homophilic MAG graphs show a decreasing likelihood of both path types as the distance between $b_u$ and $b_{mj}$ increases. If any of these paths yielded a message reaching agent $u$, then combined with the probabilities in Table 3, we see that target agents with belief strength distance 2 or less from the message will likely believe it, and update accordingly. This is exactly what we see qualitatively in the above results, as regardless of graph topology, agents with belief strength 4 or higher quickly adopt a stronger belief of 6 from the message, and agents with belief strength of 3 eventually update after enough messages reach them.

## Discussion

From our in-silico experiments, it is clear that the three types of social contagion affect populations differently given the same initial conditions and beliefs to spread. We were able to show,

as predicted, that at the individual level, simple and proportional threshold contagion methods change agent belief strengths in a manner that does not depend on what they were believing previously. In the *single* and *split* message set conditions, many agents' belief strengths were able to be swayed from value-to-value regardless of their initial belief. Thus, these contagion models do not capture the cognitive phenomena that motivated our experiments.

On the other hand, our simple individual cognitive contagion and cascade models also performed as expected. The results were fairly robust across several graph topologies. In the *single* and *split* message set conditions, most agents following the DCC function did not change their belief strength over time. This fits the underlying social theory because the messages were too far from what agents initially believed, so not updating their beliefs accurately models the defensive or entrenching effects observed when people are exposed to identity-related beliefs that they do not agree with [23, 31]. The only message set condition that was able to sway the entire population in the cognitive cascade condition was the *gradual* set.

The POD model with DCC also appears to capture population-level trends in opinion data that originally motivated our study. Results match the partisan polarization phenomena being observed [6–9, 29], as agents who only update their belief strengths in this manner are highly unlikely to be swayed by belief strengths that are too far from theirs. Once swayed in one direction or another, our agents could not adopt significantly differing belief strengths without being nudged along. Moreover, our results appear convergent with results from emerging work that use ABM with similar polar belief assumptions. Sikder et al. demonstrate in their ABM of biased agents—those only accepting congruent information—that they engender a mix of final opinions among the agents in a manner that appears robust against graph topology [72]. This result also seems to be in line with work on polarization which argues that having biased agents is key to bringing about polarized beliefs, regardless of homophilic network structure [75–77]. Though, our results are convergent under a different model: the cited works employ typical social contagion models rather than one driven by an institutional spreader leading to cascades, as ours does. The questions asked in the cited works could also be asked of cascading contagion models in future study.

Given these similarities in results, even our very simple model may be able to lend insight into potential ways to deal with spread of conspiracy beliefs—though we in no way mean for these to be taken as policy recommendations. Our model and experiments would need to be modified and parameterized to fit a given narrative spread scenario in order to be grounded enough to draw conclusions from. This may be appropriate for a follow-up study which applies this model, with more detail, to the spread of a COVID-related belief such as belief in mask-wearing.

Our analysis revealed that the three most important factors in swaying any agent's belief were (1) whether or not an agent is exposed to a message, (2) number of exposures, and (3) the difference between prior agent beliefs and those expressed in the message. Even if (1) and (2) are met, as in some attempts to debunk misinformation [5], (3) would prevent staunch conspiracy believers from changing beliefs if exposed to a contradictory message. Some analyses attempt to focus on the network structure [2, 16, 99]—i.e. (1) and (2)—without acknowledging that individual psychology is just as important—as in (3). Our model, which captures both network effects and individual effects, therefore gives novel insights into a more holistic intervention. Our analysis of results showed that for a highly homophilic network (a trait present in real social networks), certain messages have a slim chance of reaching those with certain beliefs. These theoretical results were confirmed in supplemental experiments (results in S15–S18 Figs) where graphs with high homophily and low node degree yielded smaller cascades and less contagion—even with the DCC function. The interplay between network structure and cognitive function (e.g. "stubbornness") on cascade results could benefit from further

study, as it likely has empirical analogues that are crucial to understanding reality. Any intervention would need to take these factors into account, and imagine what types of messages would be most likely to reach certain populations.

Moreover, on the individual level, an intervention under our model would have to gradually nudge the agent away from an undesirable belief strength. This brings *the individual* into the debate over interventions. The only message sets from our experiments that successfully swayed all agents in the population were those which gradually eased agents from one belief polarity to another. It should be noted that belief change tactics aimed at individuals' psychology are already widely considered and used by private and public institutions to manipulate target population beliefs, but not yet in the case of domestic misinformation [33, 100, 101]. The ethics of these interventions—often targeted at anti-radicalization, voter manipulation, or behavior change for economic gain—are clearly fraught. This begs more analysis of the ethics of intervention techniques, but such an endeavor is easily outside the scope of this report.

## Limitations and future work

Our models are in their early stages of development. While the motivation for models of cognitive cascades exists, there are several steps that still must be taken in order to flesh the model out further. For instance, in the social science literature that backs misinformation and identity-based reasoning effects, there is a great deal of evidence backing belief effects that rely on in/out-group effects [14, 22, 84], trust in message sources [10], more nuanced belief structures [31, 102–104], or effects of emotion on social contagion [14, 32]. These theories and findings could motivate more complex agent cognitive models than either the simple sigmoid distance function, or the singular proposition, we used in our model.

Our cognitive cascade model could also be made more complex in ways that would lend themselves to interesting analysis. For one, there could be added complexity when it comes to agent prior beliefs, or contagion functions. For the former, agent priors could be drawn from distributions other than the uniform distribution. Concerning the latter, parameters in the contagion functions themselves could be distributed to lead to varying levels of "gullible" versus "stubborn" agents—rendering the graph even more heterogenous. These techniques are common in opinion diffusion models [15]. Prior agent beliefs or contagion function parameters could otherwise be initialized from empirical data.

Moreover, there are potential model features that could be added to introduce a dynamic quality to the structure of the network. Some diffusion models we reviewed incorporate the ability for agents to switch information sources [72] and neighbor connections [81] based on the degree of agreement between their modeled beliefs. In media studies, however, there is no clear consensus on the behavior of individuals when it comes to choosing or switching media sources [52, 57]. Media studies dub consumer preference for bias-confirming media "selective exposure," and studies have shown both evidence of its existence [53–55] (often with small effects), and against it [8, 52, 56]. A more thorough synthesis of this literature could lead to another model layer that captures meaningful aspects of reality.

Further, a more rigorous application of the model to empirical population-level belief changes would help in verifying the legitimacy of the model's results. Using both real network structures, such as snapshots of social media networks, and real institutional message data, such as tweets or posts from notable figures, would be steps forward for this goal. While ABMs have been used to model spread of misinformation [16], social media messages [99, 105], and value-laden topics [106], it appears that few have verified outcomes against ground truth. Among other reasons, this is likely due to the fact that such data seems difficult to obtain.

There is great promise for computational social scientific tools like ABM to leverage computational power to tackle complex social problems previously limited to thought experiments and small experiments—such as modeling misinformation spread. However, if the promise is to be fulfilled, more work must be done to motivate and empirically ground both individual agent models, and global network structures and population models. But this work is a step towards establishing fruitful collaboration between the computational modeling community and social scientists in order to tackle one of the greatest political challenges of our time.

## Conclusion

This paper lays out what we call a cognitive cascade model: a combination of an individual cognitive contagion model for identity-related belief spread embedded in a *Public Opinion Diffusion* (POD) model in which external, institutional agents (modeling media companies) dictate influence of internal agent beliefs. The cognitive cascade model, by giving each individual agent a cognitive model to direct belief update, allows a level of expressiveness above existing simple and proportional threshold contagion models typically used in network cascade analysis. Moreover, adding institutional agents to drive belief cascades in an opinion diffusion model using various network topologies adds insights from network science to typical opinion diffusion studies. After proposing the cognitive cascade model, we compared potential cognitive contagion functions to arrive at one capturing misinformation spread—what we called a *Defensive Cognitive Contagion* (DCC) function—which adequately captured the cognitive dissonance and exposure effects referenced in empirical literature. This allowed us to run simulation models of networked populations of agents whose belief strength in a given proposition is influenced by an external agent. Across several graph topologies, keeping the cascade model consistent, we compared simulation results for cascades using simple contagion, proportional threshold contagion, and our DCC function. Analysis of these results revealed that our DCC function is much less sensitive to graph topology than the other cascade methods. It showed that the crucial factor in belief change was not only who surrounds a given agent but also the content of any message and its relation to the agent's prior belief. We concluded by briefly motivating potential interventions to correct misinformation and conspiracy beliefs that address the individual and the network holistically, rather than only the network they are embedded in.

## Supporting information

**S1 Text. Process for choosing contagion parameters.** An outline of our decision process for choosing simple and proportional threshold parameter values of $p = 0.15$ and $\gamma = 0.35$ for in-silico experiments.
(TEX)

**S2 Text. Process for testing different belief resolutions and analysis of results.** An outline of our process for setting up and running versions of our main contagion experiments with different belief resolutions (2, 3, 5, 9, 16, 32, and 64), including an analysis of some of the results.
(TEX)

**S3 Text. Process for comparing contagion results with differing simulation run counts.** An outline of our process for setting up, running, and analyzing versions of our main contagion experiments with different simulation run counts (10, 50, and 100).
(TEX)

**S4 Text. Process for running correlation analyses between contagion results.** An in-depth explanation of the correlation measures we used to measure similarity between contagion result data.
(TEX)

**S1 Table. Results of parameter sweeping simple contagion value $p$.** A table of timestep $t$ at which simple contagion with probability $p$ spread $b = 6$ to at least 90% of agents in different graph topologies (at a belief resolution of 7). $\emptyset$ indicates the contagion never reached at least 90% of agents.
(TEX)

**S2 Table. Correlation test results with low scores between run count combinations.** Correlation tests that yielded low scores for certain experimental combinations of graph type and message set.
(TEX)

**S1 Fig. Proportional threshold contagion variance on WS small world networks.** Proportional threshold contagion results over ten iterations of a Watts-Strogatz small world network with $N = 500$, initial neighbors $k = 5$, and rewiring chance $\rho = 0.5$. Graphs show the mean percent of agents who believe $b \in B$, color coded by $b$ value, plotted against time step. Shaded portions show variance over iterations.
(PNG)

**S2 Fig. Proportional threshold contagion variance on BA preferential attachment networks.** Proportional threshold contagion results over ten iterations of a Barabási-Albert preferential attachment network with $N = 500$, and added edges $m = 3$. Graphs show the mean percent of agents who believe $b \in B$, color coded by $b$ value, plotted against time step. Shaded portions show variance over iterations.
(PNG)

**S3 Fig. Additional results from *single* message set contagion with linear cognitive contagion functions.** The *single* message set on an ER random graph, $N = 250$, $\rho = 0.05$, for agents updating their beliefs based on the inverse linear cognitive contagion function in Eq (5) in the main paper. Graphs display percent of agents who believe $B$ with strength $b$ over time. The left graph shows agents parameterized to be "gullible" ($\gamma = 1$, $\alpha = 0$); the middle shows "normal" agents ($\gamma = 1$, $\alpha = 1$), and the right, "stubborn" agents ($\gamma = 10$, $\alpha = 20$).
(PNG)

**S4 Fig. Additional results from *single* message set contagion with threshold cognitive contagion functions.** The *single* message set on an ER random graph, $N = 250$, $\rho = 0.05$, for agents updating their beliefs based on the threshold cognitive contagion function in Eq (2) in the main paper. Graphs display percent of agents who believe $B$ with strength $b$ over time. The left graph shows agents parameterized to be "gullible" ($\gamma = 6$); the middle shows "normal" agents ($\gamma = 3$); and the right, "stubborn" agents ($\gamma = 1$).
(PNG)

**S5 Fig. Additional results from *single* message set contagion with sigmoid cognitive contagion functions.** The *single* message set on an ER random graph, $N = 250$, $\rho = 0.05$, for agents updating their beliefs based on the sigmoid cognitive contagion function in Eq (6) in the main paper. Graphs display percent of agents who believe $B$ with strength $b$ over time. The left graph shows agents parameterized to be "gullible" ($\alpha = 1$, $\gamma = 7$), the middle shows "normal" agents

($\alpha = 2$, $\gamma = 3$), and the right, "stubborn" agents ($\alpha = 4$, $\gamma = 2$).
(PNG)

**S6 Fig. Additional results from *gradual* message set contagion with linear cognitive contagion functions.** The *gradual* message set on an ER random graph, $N = 250$, $\rho = 0.05$, for agents updating their beliefs based on the inverse linear cognitive contagion function in Eq (5) in the main paper. Graphs display percent of agents who believe $B$ with strength $b$ over time. The left graph shows agents parameterized to be "gullible" ($\gamma = 1$, $\alpha = 0$); the middle shows "normal" agents ($\gamma = 1$, $\alpha = 1$), and the right, "stubborn" agents ($\gamma = 10$, $\alpha = 20$).
(PNG)

**S7 Fig. Additional results from *gradual* message set contagion with threshold cognitive contagion functions.** The *gradual* message set on an ER random graph, $N = 250$, $\rho = 0.05$, for agents updating their beliefs based on the threshold cognitive contagion function in Eq (2) in the main paper. Graphs display percent of agents who believe $B$ with strength $b$ over time. The left graph shows agents parameterized to be "gullible" ($\gamma = 6$); the middle shows "normal" agents ($\gamma = 3$); and the right, "stubborn" agents ($\gamma = 1$).
(PNG)

**S8 Fig. Additional results from *gradual* message set contagion with sigmoid cognitive contagion functions.** The *gradual* message set on an ER random graph, $N = 250$, $\rho = 0.05$, for agents updating their beliefs based on the sigmoid cognitive contagion function in Eq (6) in the main paper. Graphs display percent of agents who believe $B$ with strength $b$ over time. The left graph shows agents parameterized to be "gullible" ($\alpha = 1$, $\gamma = 7$), the middle shows "normal" agents ($\alpha = 2$, $\gamma = 3$), and the right, "stubborn" agents ($\alpha = 4$, $\gamma = 2$).
(PNG)

**S9 Fig. Contagion result comparisons using a belief resolution of 2.** The *single*, *split*, and *gradual* message sets on a Barabási-Albert preferential attachment graph, $N = 500$, and added edges $m = 3$. Graphs display percent of agents who believe some $b$ in
$B = \{b, 0 \le b \le 2\}, b \in \mathbb{Z}$.
(PNG)

**S10 Fig. Contagion result comparisons using a belief resolution of 3.** The *single*, *split*, and *gradual* message sets on a Barabási-Albert preferential attachment graph, $N = 500$, and added edges $m = 3$. Graphs display percent of agents who believe some $b$ in
$B = \{b, 0 \le b \le 3\}, b \in \mathbb{Z}$.
(PNG)

**S11 Fig. Contagion result comparisons using a belief resolution of 5.** The *single*, *split*, and *gradual* message sets on a Barabási-Albert preferential attachment graph, $N = 500$, and added edges $m = 3$. Graphs display percent of agents who believe some $b$ in
$B = \{b, 0 \le b \le 5\}, b \in \mathbb{Z}$.
(PNG)

**S12 Fig. Contagion result comparisons using a belief resolution of 9.** The *single*, *split*, and *gradual* message sets on a Barabási-Albert preferential attachment graph, $N = 500$, and added edges $m = 3$. Graphs display percent of agents who believe some $b$ in
$B = \{b, 0 \le b \le 9\}, b \in \mathbb{Z}$.
(PNG)

**S13 Fig. Contagion result comparisons using a belief resolution of 16.** The *single*, *split*, and *gradual* message sets on a Barabási-Albert preferential attachment graph, $N = 500$, and added

edges $m$ = 3. Graphs display percent of agents who believe some $b$ in
$B = \{b, 0 \leq b \leq 16\}, b \in \mathbb{Z}$.
(PNG)

**S14 Fig. Contagion result comparisons on BA preferential attachment networks using a belief resolution of 32.** The *single*, *split*, and *gradual* message sets on a Barabási-Albert preferential attachment graph, $N$ = 500, and added edges $m$ = 3. Graphs display percent of agents who believe some $b$ in $B = \{b, 0 \leq b \leq 32\}, b \in \mathbb{Z}$.
(PNG)

**S15 Fig. Contagion result comparisons on MAG networks using a belief resolution of 32.** The *single*, *split*, and *gradual* message sets on a Multiplicative Attribute Graph, $N$ = 500, and $\Theta$ generated from the same formula in Eq (8) in the main paper—i.e. one that brings about high levels of homophily so agents would rarely connect to agents more than 3 belief values away from them. Graphs display percent of agents who believe some $b$ in $B = \{b, 0 \leq b \leq 32\}, b \in \mathbb{Z}$.
(PNG)

**S16 Fig. Contagion result comparisons on WS small world networks using a belief resolution of 64.** The *single*, *split*, and *gradual* message sets on a Watts-Strogatz small world network with $N$ = 500, initial neighbors $k$ = 5, and rewiring chance $\rho$ = 0.5. Graphs display percent of agents who believe some $b$ in $B = \{b, 0 \leq b \leq 64\}, b \in \mathbb{Z}$.
(PNG)

**S17 Fig. Contagion result comparisons on BA preferential attachment networks using a belief resolution of 64.** The *single*, *split*, and *gradual* message sets on a Barabási-Albert preferential attachment graph, $N$ = 500, and added edges $m$ = 3. Graphs display percent of agents who believe some $b$ in $B = \{b, 0 \leq b \leq 64\}, b \in \mathbb{Z}$.
(PNG)

**S18 Fig. Contagion result comparisons on MAG networks using a belief resolution of 64.** The *single*, *split*, and *gradual* message sets on a Multiplicative Attribute Graph, $N$ = 500, and $\Theta$ generated from the same formula in Eq (8) in the main paper—i.e. one that brings about high levels of homophily so agents would rarely connect to agents more than 3 belief values away from them. Graphs display percent of agents who believe some $b$ in $B = \{b, 0 \leq b \leq 64\}, b \in \mathbb{Z}$.
(PNG)

**S19 Fig. Simple and proportional threshold contagion results on ER random networks.** Simple (top row) and proportional threshold (bottom row) contagion on ER random networks with $N$ = 500, and connection chance $\rho$ = 0.05. Graphs show the percent of agents who believe $B$ with strength $b$ over time.
(PNG)

**S20 Fig. Simple and proportional threshold contagion results on WS small world networks.** Simple (top row) and complex (bottom row) contagion on a Watts-Strogatz small world network with $N$ = 500, initial neighbors $k$ = 5, and rewiring chance $\rho$ = 0.5. Graphs show the percent of agents who believe $B$ with strength $b$ over time. Asterisks (*) denote these contagions had significant variance over simulation iterations.
(PNG)

**S21 Fig. Simple and proportional threshold contagion results on BA preferential attachment networks.** Simple (top row) and proportional threshold (bottom row) contagion on

Barabási-Albert preferential attachment networks with $N = 500$, and added edges $m = 3$. Graphs show the percent of agents who believe $B$ with strength $b$ over time. Asterisks ($*$) denote these contagions had significant variance over simulation iterations.
(PNG)

**S22 Fig. Simple and proportional threshold contagion results on homophilic MAG networks.** Simple (top row) and proportional threshold (bottom row) contagion on a homophilic MAG networks with $N = 500$, and $\Theta_b$ detailed in Eq (8). Graphs show the percent of agents who believe $B$ with strength $b$ over time.
(PNG)

**S23 Fig. Graphical contagion results for low correlation combinations across simulation run counts.** Graphical results of contagion cascades across 10, 50, and 100 simulation runs for specific graph-message set combinations. Results displayed are those which yielded the lowest correlation scores between simulation run counts.
(PNG)

## Author Contributions

**Conceptualization:** Nicholas Rabb, Jan P. de Ruiter, Matthias Scheutz.

**Funding acquisition:** Lenore Cowen, Matthias Scheutz.

**Methodology:** Nicholas Rabb, Lenore Cowen, Jan P. de Ruiter, Matthias Scheutz.

**Software:** Nicholas Rabb.

**Supervision:** Lenore Cowen, Matthias Scheutz.

**Visualization:** Nicholas Rabb.

**Writing – original draft:** Nicholas Rabb, Lenore Cowen, Jan P. de Ruiter, Matthias Scheutz.

**Writing – review & editing:** Nicholas Rabb, Lenore Cowen, Jan P. de Ruiter, Matthias Scheutz.

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
