## [Decision Letter · Decision Letter 0]

18 Jun 2021

PONE-D-21-06938

Cognitive contagion: How to model (and potentially counter) the spread of fake news

PLOS ONE

Dear Dr. Rabb,

Thank you for submitting your manuscript to PLOS ONE. Your manuscript has been carefully reviewed and overall regarded as a valuable contribution by the referees. I personally share the appreciation expressed by the referees and consider this work a valuable contribution to a class of models whose importance is raising and that are still open to many important development. However, some important aspects in need of further consideration have been commented. Therefore, we invite you to submit a revised version of the manuscript that addresses the points raised during the review process.

Among the others, some comments need particular consideration and should be thoroughly addressed in the revised version of the manuscript: 

Referee 2 and Referee 3 provided specific and detailed comments about several important references useful to better contextualize your work and make more accurate statements about its contribution;Referee 3 provided detailed comments (see attached file) on agent-based modelling that could greatly improve the analysis and presentation of the work.

We look forward to receiving your revised manuscript.

Kind regards,

Marco Cremonini, Ph.D.

University of Milan

Academic Editor

PLOS ONE

Journal Requirements:

2. Please reupload your manuscript as a .docx or editable .pdf file

Reviewers' comments:

Reviewer's Responses to Questions

**Comments to the Author**

1. Is the manuscript technically sound, and do the data support the conclusions?

Reviewer #1: Yes

Reviewer #2: Yes

Reviewer #3: Partly

2. Has the statistical analysis been performed appropriately and rigorously? 

Reviewer #1: Yes

Reviewer #2: Yes

Reviewer #3: Yes

3. Have the authors made all data underlying the findings in their manuscript fully available?

Reviewer #1: Yes

Reviewer #2: Yes

Reviewer #3: Yes

4. Is the manuscript presented in an intelligible fashion and written in standard English?

Reviewer #1: Yes

Reviewer #2: Yes

Reviewer #3: Yes

5. Review Comments to the Author

Reviewer #1: The paper puts forward an agent-based model of opinion dynamics. The main innovation of the paper lies in its assumptions on the behaviour of the agents. Namely, in addition to the mechanistic aspects of information diffusion that are traditionally incorporated in most opinion dynamics models, the paper also makes specific assumptions on the behaviour of the agents, and explicitly models their likelihood to be receptive to new information depending on their current beliefs. This assumption is aimed at modelling phenomena of collective misinformation, such as the refusal to believe in the reality of covid-19.

I find the paper excellent in every possible way. I really like this modelling approach, and I think it's the only one with some hope of yielding empirically testable predictions, something which is sorely lacking in most of the opinion dynamics literature. I think the results presented by the authors are convincing, and I find their analysis (pages 19-22) very thorough and instructive. I basically have no criticisms or major comments, and I believe the paper should be published in a form very close to its current one. I only have a couple of minor suggestions, listed in the following:

1. The only aspects of the model I didn't find entirely convincing are related to its assumptions on institutional agents (let's call them news sources) and their relationships with their subscribers. Is it realistic to assume that a subscriber should keep listening to (as in maintaining their link with) a news source he/she consistently doesn't agree with? If I understand this aspect of the model correctly, it seems that non-institutional agents are not allowed to re-evaluate their links to news sources. Could this be accommodated, e.g., by expanding the number of news sources in order to simulate the effects of a competitive news market? I fully realise these points are probably way beyond the scope of the current paper, but I still think it could be interesting to see them acknowledged/discussed in the final section of the paper.

2. Have the authors considered the scenario in which the parameters in Eq. (6) are drawn from a suitable distribution? I think this would be a very interesting extension of the model towards a more realistic description of a heterogeneous audience. Again, like in the previous point I don't think it's necessary to include additional results in this respect, and I only recommend to include this aspect as a point of discussion.

3. I think the authors should take a look at Sikder et al., "A minimalistic model of bias, polarization and misinformation in social networks", Scientific Reports (2020). As they will see, that paper starts from very similar premises to those of their study, and also makes explicit assumptions about the behaviour of agents with respect to new information depending on their beliefs. The two models are ultimately quite different and focus on different aspects, but it's quite interesting to see how both achieve remarkable consistency in their results across very different network topologies (see, e.g., Fig. 2 in the Scientific Reports paper).

Reviewer #2: Dear Nicholas, Lenore, Jan, and Matthias,

Thank you for giving me the opportunity to review your paper “Cognitive contagion: How to model (and potentially counter) the spread of fake news”. You are tackling an important and timely topic with a rigorous approach, and yield an insightful conclusion. You point out (correctly, in my view) that traditional models of social contagion generally don’t account for the internal state of the individual, and fail to consider how this internal state influences their adoption decisions. With some minor modifications, I would like to see your paper in print.

pg 4, eqn 1 (and other equations describing changes in beliefs)

- I tripped up here at first because I didn’t recognize that this was an equation describing changes to the state at `t+1` as a function of the state at `t`. It might clarify things include a subscript for the timestep in these equations, ie: p(b_{u,t+1} = … | b_{u,t}) or p(b_{u}(t+1) = … | b_{u}(t). Also here, can you clarify in the text that ‘u’ is the focal adopter and ‘v’ is the focal exposer?

Pg 5: Complex contagion

There is some ambiguity in the literature about the precise meaning of “complex contagion”, and how it captures the need for social reinforcement.

Kempe, Kleinberg and Tardos (2003) (https://www.cs.cornell.edu/home/kleinber/kdd03-inf.pdf) articulate the difference between a (proportional) threshold model, and an independent cascade model (which most folks would call ‘simple contagion’). Their description of a threshold model is what you (and a lot of other papers) describe as complex contagion.

Centola and Macy actually have a different requirement for “complex” contagion than either mentioned by KKT: that a minimum absolute number (not fraction!) of individuals need to expose you to the belief before you adopt. This is important to their argument because they are making the claim that small world networks don’t lead to fast diffusion unless you have “wide bridges” across the network. (Of course, there are some new papers suggesting this falls apart if you have a stochastic decision rule, so maybe we should take it with a grain of salt.)

- I think if you want to use the term “complex contagion”, that’s probably ok, (given the ambiguity in common usage) just note that you’re using the proportional threshold interpretation, not the strict requirement from the Centola and Macy paper, which you’re citing at the moment. Even better, contribute to making the term less ambiguous by differentiating between Centola’s complex contagion and what’s in the Schelling model.

- The other place it will be relevant is in your section on complex contagion in WS networks (pg 15). You say that “interestingly, complex contagion was successful on the WS graphs”. I’m guessing that this is interpreted as surprising because of the Centola and Macy finding that complex contagion should be slower than simple contagion in small-world networks. I think the resolution is that you’re using a different interpretation of complex contagion (and that WS networks are not degree regular). I would suggest you choose one of the two usages and stick with it throughout.

Pg. 6: Cognitive contagion model

DeGroot 1974 (https://www.jstor.org/stable/2285509) has a classic model of updating degrees of belief in response to neighbors beliefs, with some weighting on neighbors. Your model could be considered an extension of the DeGroot model that makes the weights on individuals an increasing function of difference between individuals, ie. accounting for homophily. In this vein, you probably also want to look at Dandekar, Goel and Lee 2013 (https://www.pnas.org/content/110/15/5791) and see whether you agree with their conclusions, and if not, why not.

Axelrod 1997 (https://journals.sagepub.com/doi/abs/10.1177/0022002797041002001) has a model that accounts for individuals paying more attention to similar individuals, as does DellaPosta 2015 (https://www.jstor.org/stable/10.1086/681254) and Baldassarri and Bearman (https://www.jstor.org/stable/25472492). In these cases, homophily is conceptualized as similarity on other belief statements, rather than just on a single belief. The homophily element of their work is similar to what you’re doing here with a single belief. If homophily itself is the driving factor, your simulations should give the same results. If it’s about gradual belief updating, you may get different results.

There are a couple folks looking at within-person interactions between beliefs (a third way to conceptualize the interaction between existing beliefs and adoption decisions!). Goldberg and Stein 2018 (https://journals.sagepub.com/doi/abs/10.1177/0003122418797576) is a good start, as is Friedkin et al 2016 (https://science.sciencemag.org/content/354/6310/321/tab-figures-data) - although Friedkin’s model suffers from an ambiguity in whether the outcome is due to social contagion or to the assumed logic constraints. (I’ve done some work on this too (https://arxiv.org/abs/2010.02188) but please don’t read this as grubbing for citations - if you want to cite someone on those ideas, go with the folks above…)

- Please check that your results are insensitive to the number of levels in your model. 7 points is arbitrary (which is fine) but we don’t expect it to be a faithful representation of what goes on in people’s heads (regardless of what Likert thinks). Sometimes these things matter - just make sure that it doesn’t matter here. Make a 100-point model or something that approximates a continuous scale, just to be sure. You can put it in the supplement, and put a note in the main body to say that you checked.

- Another consequential parameter choice is the ’stubbornness’ of individuals. You do a good job exploring different options here, but your justification for why the stubborn parameters are the ones you carry forward is grounded in the outcomes you want to see. Again, it’s not a problem to do so if the purpose of the model is to highlight a possible outcome and describe when it is likely to occur. But, it does seem a bit like sampling on the dependent variable. Can you either justify why we should expect this to be the right choice without reference to the simulation outcomes (you could use a micro-level analysis with a single agent) or just make this assumption explicit? Something like “sometimes, people are stubborn. When this is true, here’s what we expect to happen”. Then you’re just making a micro-level assumption and exploring its consequences, rather than trying to say “the world is this way…”.

- I’m surprised 10 simulation runs is enough to get a stable result. I usually end up increasing the number of runs (10, 20, 50, 100, 200, etc) until I don’t see any difference in the resulting averages, and then do 2-10 times as many as that. If you have done thousands and found the same results, you can say that you did them, but that the results only show the results of 10 because the effects are so robust. If you haven’t done a large number, it might be a good idea, as they aren’t expensive.

- Do you pick a random agent to be the broadcaster, or are they different entities in your model? I had trouble working that out.

- Fig 3. beliefs don’t match what’s in the text (in the text, u is 1, but they are exposed to 6, in the fig, they believe 6 but are exposed to 0). This had me confused for a bit as I wasn’t sure I was reading the figure correctly.

Pg. 9: Contagion experiments

- I’m aware that the term “experiments” is sometimes used to describe running simulations under different conditions, and in the broadest sense (trying something to see what happens) it’s an appropriate term. However, I do feel that as a community it would be useful to distinguish between computer-assisted “gedanken-experiments” and experiments that make manipulations in a lab or in the real world with human participants. The first is for theory building - an essential part of the scientific division of labor - and the second for theory testing. I certainly won’t twist your arm, but you may find it helpful to be more explicit that you are exploring the macro-level consequences of a micro-level assumption, using a simulation to overcome the brain’s inability to see these consequences on its own. This will clarify your contribution for readers, as they’ll know what to expect in the next section.

- Do your behavior over time charts (Fig 6. - 16.) show a true T0? I.e. the distribution of individual beliefs *before* any adoption has taken place? I would imagine that I should see the T0 for all charts to be identical (same starting conditions) and fairly similar to the bottom charts in Fig 9. If I’m not mistaken, please include those t0 belief distributions in your plots - it would be helpful to know where individuals are coming from.

pg 14. Section on comparing contagion models

- You vary social network structure and then describe the qualitative differences of your model in each of these graphs. We know that different network structures yields different shaped diffusion curves, and so this isn’t your point. Instead, I believe you are suggesting that the differences between network structures under the traditional contagion models are not the same as the differences under your own model. This is a pretty complicated comparison to make, and as the text currently reads, I personally (with a reviewer’s normal cognitive impairments…) have trouble understanding what I should take away. Specifically, I’m having a hard time distinguishing the effect of your model from the effect of the changes to the network structure, in this section in particular. One idea might be to make a table with different adoption rules on one axis, and different social networks on the other, and for a specific metric compare the differences across condition. Then you can highlight why your model changes our expectation about the effect of network structure. You could have multiple tables for different metrics.

pg. 19 Section on Analysis of Results

- Your analysis suggests that because there is almost always a path for information to reach an individual from the broadcast node, then the internal logic of an individual’s decision rule is more determinant of the outcome than the social network structure (if I understand it correctly!). A good comparison to make might be a case where there is no social influence at all, essentially an individual adoption model from the “broadcast” source, i.e. a star network with the broadcast source at the center. Then you can compare the effects of the other network structures to see which components of the outcome are due to the characteristics exemplified in each of the networks (clustering, short path lengths, high/low degree, etc).

The challenge here is that this sets up a horse race between the effects of individual cognition, and the effects of network structure. You’re not in a position to really adjudicate between these two effects, as in your simulation the relative magnitudes will be entirely dependent on the parameters of “stubbornness” that get selected in the model. You could turn this into an opportunity, however, by describing (qualitatively) the conditions under which we should expect one of the effects to dominate over the other. Then you get to say something like “and so an important piece of empirical research will be to determine which of the two regimes (the types of social contagion we care about) fall into”.

Pg. 22 Discussion

- I appreciate your disclaimer that you are not making policy recommendations based on your results. It is fully appropriate to acknowledge that you are building theory that we as a community should subject to test before using it to derive policy. I also understand how (academically) we feel pressure to say that our work is “policy relevant”. Unfortunately, I think our community has a tendency to try and have it both ways, and try and both make policy claims and then hedge them at the same time, and it’s never really that clean. I think you may find it more comfortable to move fully away from the “policy recommendation” language, and instead describe how your work sets up what is arguably a very interesting follow-on study to confirm your understanding. (Plos one gives you this luxury - take advantage of it!) Then your description of an intervention can be wholeheartedly about an intervention in an experimental context, without the need for the disclaimer. You can describe how an experiment would differ from individual-level behavioral interventions in the psychology literature, and what that would add to our understanding. Best yet, you’d have your next paper lined up nicely.

Thank you again for reading through what has become an inexcusably long review. I’m certain that you can address the questions I have without too much effort, and I look forward to seeing the revisions.

James Houghton

Reviewer #3: This paper presents a model of diffusion that is grounded in the cognition of each agents. By comparing different graph types to the exposure to information/beliefs coming from an institutional source, the authors try to show how adding complexity to the definition of an agent may lead to map more closely what happened, for example, with beliefs around COVID-19.

While certainly interesting and timely, the article presents a series of concerns that make the message a bit less effective than it could be. The most relevant concerns I had are those around (a) the claim that more complex agents is something “new” of this model, while the ABM community has been discussing it since its beginnings, (b) the use of ABM (especially the “why” ABM), (c) the complete lack of reference to description and reporting standards in presenting the ABM, and (d) the report of results shows that you are probably trying to do too much with just one paper.

I have detailed what I mean by this in the attached file, I hope you find them useful. I enjoyed reading your paper. Best of luck with your research!

6. PLOS authors have the option to publish the peer review history of their article (what does this mean?). If published, this will include your full peer review and any attached files.

Reviewer #1: **Yes: **Giacomo Livan

Reviewer #2: **Yes: **James Houghton

Reviewer #3: **Yes: **Davide Secchi

---

## [Author Response · Author response to Decision Letter 0]

25 Aug 2021

We want to thank all three reviewers for comments and suggested revisions and references that greatly improved our revised draft of the paper. We would like to point out, that some comments reviewer 2 and 3, have led us to decide to change our title, and a main term in the paper: in particular, we now are calling our method “cognitive cascade” instead of “cognitive contagion” (as they have pointed us to literature that also uses simple cognitive models in agents) and have now renamed some of the contagion models we compare against to be more consistent with terminology of the past work (see comments to reviewer 2 below, in regards to “complex” versus proportional threshold contagion models). More detailed comments to individual reviewers appear next. 

REVIEWER 1

Reviewer #1: The paper puts forward an agent-based model of opinion dynamics. The main innovation of the paper lies in its assumptions on the behaviour of the agents. Namely, in addition to the mechanistic aspects of information diffusion that are traditionally incorporated in most opinion dynamics models, the paper also makes specific assumptions on the behaviour of the agents, and explicitly models their likelihood to be receptive to new information depending on their current beliefs. This assumption is aimed at modelling phenomena of collective misinformation, such as the refusal to believe in the reality of covid-19.

I find the paper excellent in every possible way. I really like this modelling approach, and I think it's the only one with some hope of yielding empirically testable predictions, something which is sorely lacking in most of the opinion dynamics literature. I think the results presented by the authors are convincing, and I find their analysis (pages 19-22) very thorough and instructive. I basically have no criticisms or major comments, and I believe the paper should be published in a form very close to its current one. I only have a couple of minor suggestions, listed in the following:

==== OUR RESPONSE ====

Thank you very much for the kind words!

==== REVIEWER ====

1. The only aspects of the model I didn't find entirely convincing are related to its assumptions on institutional agents (let's call them news sources) and their relationships with their subscribers. Is it realistic to assume that a subscriber should keep listening to (as in maintaining their link with) a news source he/she consistently doesn't agree with? If I understand this aspect of the model correctly, it seems that non-institutional agents are not allowed to re-evaluate their links to news sources. Could this be accommodated, e.g., by expanding the number of news sources in order to simulate the effects of a competitive news market? I fully realise these points are probably way beyond the scope of the current paper, but I still think it could be interesting to see them acknowledged/discussed in the final section of the paper.

==== OUR RESPONSE ====

We agree that a more realistic model would include a more complex relationship between how individuals link to new sources, and allow multiple news sources and include switching dynamics. We have now added the requested paragraph (along with some relevant references we found to cite) in the discussion.

==== REVIEWER ====

2. Have the authors considered the scenario in which the parameters in Eq. (6) are drawn from a suitable distribution? I think this would be a very interesting extension of the model towards a more realistic description of a heterogeneous audience. Again, like in the previous point I don't think it's necessary to include additional results in this respect, and I only recommend to include this aspect as a point of discussion.

==== OUR RESPONSE ====

 We added language advocating for this as a potential extension of the model in our discussion section:

“Our cognitive contagion could also be made more complex in ways that would lend themselves to interesting analysis. For one, there could be added complexity when it comes to agent prior beliefs, or cognitive contagion functions. For the former, agent priors could be drawn from distributions other than the uniform distribution. Concerning the latter, parameters in the contagion functions themselves could be distributed to lead to varying levels of ``gullible'' versus ``stubborn'' agents -- rendering the graph even more heterogenous. These techniques are common in opinion diffusion models (Zhang & Vorobeychik, 2019). Prior agent beliefs or contagion function parameters could otherwise be initialized from empirical data.”

==== REVIEWER ====

3. I think the authors should take a look at Sikder et al., "A minimalistic model of bias, polarization and misinformation in social networks", Scientific Reports (2020). As they will see, that paper starts from very similar premises to those of their study, and also makes explicit assumptions about the behaviour of agents with respect to new information depending on their beliefs. The two models are ultimately quite different and focus on different aspects, but it's quite interesting to see how both achieve remarkable consistency in their results across very different network topologies (see, e.g., Fig. 2 in the Scientific Reports paper).

==== OUR RESPONSE ====

This is a great reference -- thank you for the suggestion. We now cite this in both the end of the introduction and in discussion and agree that it is very related and relevant work. 

REVIEWER 2

Reviewer #2: Dear Nicholas, Lenore, Jan, and Matthias,

Thank you for giving me the opportunity to review your paper “Cognitive contagion: How to model (and potentially counter) the spread of fake news”. You are tackling an important and timely topic with a rigorous approach, and yield an insightful conclusion. You point out (correctly, in my view) that traditional models of social contagion generally don’t account for the internal state of the individual, and fail to consider how this internal state influences their adoption decisions. With some minor modifications, I would like to see your paper in print.

==== OUR RESPONSE ====

Thank you for the kind words!

==== REVIEWER ====

pg 4, eqn 1 (and other equations describing changes in beliefs)

- I tripped up here at first because I didn’t recognize that this was an equation describing changes to the state at `t+1` as a function of the state at `t`. It might clarify things include a subscript for the timestep in these equations, ie: p(b_{u,t+1} = … | b_{u,t}) or p(b_{u}(t+1) = … | b_{u}(t). Also here, can you clarify in the text that ‘u’ is the focal adopter and ‘v’ is the focal exposer?

==== OUR RESPONSE ====

We agree that this suggested change in notation would improve clarity. We updated equations (1), (3), (4), (5), (6), and (7) accordingly, and added some clarifying language when the terms are introduced before equation (1).

==== REVIEWER ====

Pg 5: Complex contagion

There is some ambiguity in the literature about the precise meaning of “complex contagion”, and how it captures the need for social reinforcement.

Kempe, Kleinberg and Tardos (2003) (https://www.cs.cornell.edu/home/kleinber/kdd03-inf.pdf) articulate the difference between a (proportional) threshold model, and an independent cascade model (which most folks would call ‘simple contagion’). Their description of a threshold model is what you (and a lot of other papers) describe as complex contagion.

Centola and Macy actually have a different requirement for “complex” contagion than either mentioned by KKT: that a minimum absolute number (not fraction!) of individuals need to expose you to the belief before you adopt. This is important to their argument because they are making the claim that small world networks don’t lead to fast diffusion unless you have “wide bridges” across the network. (Of course, there are some new papers suggesting this falls apart if you have a stochastic decision rule, so maybe we should take it with a grain of salt.)

- I think if you want to use the term “complex contagion”, that’s probably ok, (given the ambiguity in common usage) just note that you’re using the proportional threshold interpretation, not the strict requirement from the Centola and Macy paper, which you’re citing at the moment. Even better, contribute to making the term less ambiguous by differentiating between Centola’s complex contagion and what’s in the Schelling model.

- The other place it will be relevant is in your section on complex contagion in WS networks (pg 15). You say that “interestingly, complex contagion was successful on the WS graphs”. I’m guessing that this is interpreted as surprising because of the Centola and Macy finding that complex contagion should be slower than simple contagion in small-world networks. I think the resolution is that you’re using a different interpretation of complex contagion (and that WS networks are not degree regular). I would suggest you choose one of the two usages and stick with it throughout.

==== OUR RESPONSE ====

We agree that it would be valuable to tease apart better in our description these two types of contagion. . We added language in several places to address this:- In the second paragraph of the Simple Contagion section we now write: “There are two popular types of social contagion models used in ABMs: simple -- also called independent cascade -- complex contagion -- which has a proportional and absolute variation.”

In the first paragraph of the Complex Contagion section: “There are two major variations of complex contagion: what is typically called a proportional threshold contagion, and what we call an absolute threshold contagion. Proportional threshold contagion creates some $\\alpha$ proportion of neighbors which must believe something for the ego agent $u$ to believe it. Absolute threshold contagion, on the other hand, may imagine some whole number $\\eta$ of neighbors who must believe something in order for the ego $u$ to believe it (Centola & Macy 2007; Granovetter,1978). We choose to use the proportional threshold model for our examples and in-silico experiments.”

Additionally, we changed our use of “complex contagion” throughout the paper to “proportional threshold contagion.”

==== REVIEWER ====

Pg. 6: Cognitive contagion model

DeGroot 1974 (https://www.jstor.org/stable/2285509) has a classic model of updating degrees of belief in response to neighbors beliefs, with some weighting on neighbors. Your model could be considered an extension of the DeGroot model that makes the weights on individuals an increasing function of difference between individuals, ie. accounting for homophily. In this vein, you probably also want to look at Dandekar, Goel and Lee 2013 (https://www.pnas.org/content/110/15/5791) and see whether you agree with their conclusions, and if not, why not.

==== OUR RESPONSE ====

We now cite these valuable references. In addition, we added a new section to the background, called Cognitive Contagion, outlining contributions from these models. We also added language in the discussion to comment on how our results align with others, kindly provided by the reviewers. 

==== REVIEWER ====

Axelrod 1997 (https://journals.sagepub.com/doi/abs/10.1177/0022002797041002001) has a model that accounts for individuals paying more attention to similar individuals, as does DellaPosta 2015 (https://www.jstor.org/stable/10.1086/681254) and Baldassarri and Bearman (https://www.jstor.org/stable/25472492). In these cases, homophily is conceptualized as similarity on other belief statements, rather than just on a single belief. The homophily element of their work is similar to what you’re doing here with a single belief. If homophily itself is the driving factor, your simulations should give the same results. If it’s about gradual belief updating, you may get different results.

There are a couple folks looking at within-person interactions between beliefs (a third way to conceptualize the interaction between existing beliefs and adoption decisions!). Goldberg and Stein 2018 (https://journals.sagepub.com/doi/abs/10.1177/0003122418797576) is a good start, as is Friedkin et al 2016 (https://science.sciencemag.org/content/354/6310/321/tab-figures-data) - although Friedkin’s model suffers from an ambiguity in whether the outcome is due to social contagion or to the assumed logic constraints. (I’ve done some work on this too (https://arxiv.org/abs/2010.02188) but please don’t read this as grubbing for citations - if you want to cite someone on those ideas, go with the folks above…)

Again, thanks for the treasure trove of helpful references which we now cite. Definitely helps to put our work in better context. 

==== REVIEWER ====

- Please check that your results are insensitive to the number of levels in your model. 7 points is arbitrary (which is fine) but we don’t expect it to be a faithful representation of what goes on in people’s heads (regardless of what Likert thinks). Sometimes these things matter - just make sure that it doesn’t matter here. Make a 100-point model or something that approximates a continuous scale, just to be sure. You can put it in the supplement, and put a note in the main body to say that you checked.

As suggested, we have conducted further experiments to ensure that the behavior of the cascades and contagion were similar for different resolutions of belief: if b was divided into 2 discrete points, 3, 5, 7, 9, 16, 32, and 64. We find the results are stable up until 16, but once the model is closer to continuous (even 64), we find that our way of modeling news sources is overwhelmed and we don’t see the same dynamics. We included resultant graphs in the Supplemental Materials (S11 Text & S12-S21 Fig) and made note of them in the main text: “We additionally ran our later in-silico experiments with lower and higher ``resolutions'' of belief: with $b$ able to take integer values between 0 and 2, 3, 5, 7, 9, 16, 32, and 64. Results from those belief resolutions are shown in S12-S21 Figs.”

- Another consequential parameter choice is the ’stubbornness’ of individuals. You do a good job exploring different options here, but your justification for why the stubborn parameters are the ones you carry forward is grounded in the outcomes you want to see. Again, it’s not a problem to do so if the purpose of the model is to highlight a possible outcome and describe when it is likely to occur. But, it does seem a bit like sampling on the dependent variable. Can you either justify why we should expect this to be the right choice without reference to the simulation outcomes (you could use a micro-level analysis with a single agent) or just make this assumption explicit? Something like “sometimes, people are stubborn. When this is true, here’s what we expect to happen”. Then you’re just making a micro-level assumption and exploring its consequences, rather than trying to say “the world is this way…”.

==== OUR RESPONSE ====

We have now added language in the second to last paragraph of the “Sigmoid Function” subsection of “Motivating Choice of $\\beta$ Functions” in response to this critique as follows: “These agents act in a way akin to what we observe from cognitive literature (Swire-Thompson et al., 2020; Festinger, 1957), albeit in a highly simplified manner: an agent who ``strongly disbelieves'' in something like COVID mask-wearing will likely only be swayed by a message that ``disbelieves'' or is ``uncertain'' about the belief. On the individual level, a maximum two relative magnitudes of belief separation, with decreasing probabilities as distance increases, seems to qualitatively match empirical work. In our simulations using more than 7 points on a belief spectrum, this argument can still be held by setting the equivalent belief ``markers'' along the spectrum, and using those to scale the contagion function.”

==== REVIEWER ====

- I’m surprised 10 simulation runs is enough to get a stable result. I usually end up increasing the number of runs (10, 20, 50, 100, 200, etc) until I don’t see any difference in the resulting averages, and then do 2-10 times as many as that. If you have done thousands and found the same results, you can say that you did them, but that the results only show the results of 10 because the effects are so robust. If you haven’t done a large number, it might be a good idea, as they aren’t expensive.

==== OUR RESPONSE ====

To answer this point, we repeated all the in-silico experiments in the “Comparing Contagion Methods” section using 50 and 100 iterations, instead of 10. As expected, we found that results did not change significantly. We now include these results in the Supplemental materials (S22 Text, S23 Table, and S24 Fig). We also added text acknowledging this in the second to last paragraph in the “Experiment Design” subsection of the main text: “We display results aggregated over 10 simulations here, with results from 50 and 100 simulations and justification for using 10 in S22 Text, S23 Table, and S24 Fig.”

==== REVIEWER ====

- Do you pick a random agent to be the broadcaster, or are they different entities in your model? I had trouble working that out.

==== OUR RESPONSE ====

We agree that this was confusing in the original draft. We now clarify that “There is a separate set of institutional agents $I$ -- entirely different entities in the ontology of our model -- which have directed edges to a set of ``subscribers’’ $S \\subseteq V$...”

==== REVIEWER ====

- Fig 3. beliefs don’t match what’s in the text (in the text, u is 1, but they are exposed to 6, in the fig, they believe 6 but are exposed to 0). This had me confused for a bit as I wasn’t sure I was reading the figure correctly.

==== OUR RESPONSE ====

We have modified the text and hopefully now made it less confusing as follows: : ”Perhaps an agent $u$ who strongly believes the proposition ($b_u = 6$) will not switch immediately to strongly disbelieving it without passing through an intermediary step of uncertainty. Given a neighbor $v$ sharing belief $b_v = 0$, agent $u$ should not adopt this belief strength, because the difference in belief strengths is clearly greater than $\\gamma$. Simple contagion would fall short because agent $u$ may simply randomly become ``infected'' with belief strength 0 by $v$ with some probability $p$. A proportional threshold contagion would similarly falter if agent $u$ were entirely surrounded by alters with belief strength 0.”

==== REVIEWER ====

Pg. 9: Contagion experiments

- I’m aware that the term “experiments” is sometimes used to describe running simulations under different conditions, and in the broadest sense (trying something to see what happens) it’s an appropriate term. However, I do feel that as a community it would be useful to distinguish between computer-assisted “gedanken-experiments” and experiments that make manipulations in a lab or in the real world with human participants. The first is for theory building - an essential part of the scientific division of labor - and the second for theory testing. I certainly won’t twist your arm, but you may find it helpful to be more explicit that you are exploring the macro-level consequences of a micro-level assumption, using a simulation to overcome the brain’s inability to see these consequences on its own. This will clarify your contribution for readers, as they’ll know what to expect in the next section.

==== OUR RESPONSE ====

Thank you for the suggestion -- we agree. We changed the term “experiment” to “in-silico experiment” throughout the text in appropriate areas, notably in the section header “In-Silico Contagion Experiments.”

==== REVIEWER ====

- Do your behavior over time charts (Fig 6. - 16.) show a true T0? I.e. the distribution of individual beliefs *before* any adoption has taken place? I would imagine that I should see the T0 for all charts to be identical (same starting conditions) and fairly similar to the bottom charts in Fig 9. If I’m not mistaken, please include those t0 belief distributions in your plots - it would be helpful to know where individuals are coming from.

==== OUR RESPONSE ====

This is a very useful suggestion. We updated all of our graphs to display the equivalent of 10 time steps of initial belief distribution before t0 begins to change them. We think this is substantially clearer. 

==== REVIEWER ====

pg 14. Section on comparing contagion models

- You vary social network structure and then describe the qualitative differences of your model in each of these graphs. We know that different network structures yields different shaped diffusion curves, and so this isn’t your point. Instead, I believe you are suggesting that the differences between network structures under the traditional contagion models are not the same as the differences under your own model. This is a pretty complicated comparison to make, and as the text currently reads, I personally (with a reviewer’s normal cognitive impairments…) have trouble understanding what I should take away. Specifically, I’m having a hard time distinguishing the effect of your model from the effect of the changes to the network structure, in this section in particular. One idea might be to make a table with different adoption rules on one axis, and different social networks on the other, and for a specific metric compare the differences across condition. Then you can highlight why your model changes our expectation about the effect of network structure. You could have multiple tables for different metrics.

==== OUR RESPONSE ====

We agree that the takeaway at the end of the comparison was confusing. We have added text in the “Analysis of Results” section under a subsection called “Belief Contagion” which lays out a statistical analysis of the results. Details about the analysis can be found in Supplemental Materials (S25 Text).

==== REVIEWER ====

pg. 19 Section on Analysis of Results

- Your analysis suggests that because there is almost always a path for information to reach an individual from the broadcast node, then the internal logic of an individual’s decision rule is more determinant of the outcome than the social network structure (if I understand it correctly!). A good comparison to make might be a case where there is no social influence at all, essentially an individual adoption model from the “broadcast” source, i.e. a star network with the broadcast source at the center. Then you can compare the effects of the other network structures to see which components of the outcome are due to the characteristics exemplified in each of the networks (clustering, short path lengths, high/low degree, etc).

The challenge here is that this sets up a horse race between the effects of individual cognition, and the effects of network structure. You’re not in a position to really adjudicate between these two effects, as in your simulation the relative magnitudes will be entirely dependent on the parameters of “stubbornness” that get selected in the model. You could turn this into an opportunity, however, by describing (qualitatively) the conditions under which we should expect one of the effects to dominate over the other. Then you get to say something like “and so an important piece of empirical research will be to determine which of the two regimes (the types of social contagion we care about) fall into”.

==== OUR RESPONSE ====

We agree with this framing, and have added language in our discussion to address it at the end of the second to last paragraph: “These theoretical results were confirmed in supplemental experiments where graphs with high homophily and low node degree yielded less contagion -- even with the DCC function. The interplay between network structure and cognitive function (e.g. ``stubbornness'') on contagion results could benefit from further study, as it likely has empirical analogues that are crucial to understanding reality.”

==== REVIEWER ====

Pg. 22 Discussion

- I appreciate your disclaimer that you are not making policy recommendations based on your results. It is fully appropriate to acknowledge that you are building theory that we as a community should subject to test before using it to derive policy. I also understand how (academically) we feel pressure to say that our work is “policy relevant”. Unfortunately, I think our community has a tendency to try and have it both ways, and try and both make policy claims and then hedge them at the same time, and it’s never really that clean. I think you may find it more comfortable to move fully away from the “policy recommendation” language, and instead describe how your work sets up what is arguably a very interesting follow-on study to confirm your understanding. (Plos one gives you this luxury - take advantage of it!) Then your description of an intervention can be wholeheartedly about an intervention in an experimental context, without the need for the disclaimer. You can describe how an experiment would differ from individual-level behavioral interventions in the psychology literature, and what that would add to our understanding. Best yet, you’d have your next paper lined up nicely.

==== OUR RESPONSE ====

We appreciate this suggestion, and have slightly changed our language in this section (in the same paragraph you comment on) as follows: “Our model and experiments would need to be modified and parameterized to fit a given narrative spread scenario in order to be grounded enough to draw conclusion from. This may be appropriate for a follow-up study which applies this model, with more detail, to the spread of a COVID-related belief such as belief in mask-wearing.”

==== REVIEWER ====

Thank you again for reading through what has become an inexcusably long review. I’m certain that you can address the questions I have without too much effort, and I look forward to seeing the revisions.

James Houghton

REVIEWER 3

Reviewer #3: This paper presents a model of diffusion that is grounded in the cognition of each agents. By comparing different graph types to the exposure to information/beliefs coming from an institutional source, the authors try to show how adding complexity to the definition of an agent may lead to map more closely what happened, for example, with beliefs around COVID-19.

While certainly interesting and timely, the article presents a series of concerns that make the message a bit less effective than it could be. The most relevant concerns I had are those around (a) the claim that more complex agents is something “new” of this model, while the ABM community has been discussing it since its beginnings, (b) the use of ABM (especially the “why” ABM), (c) the complete lack of reference to description and reporting standards in presenting the ABM, and (d) the report of results shows that you are probably trying to do too much with just one paper.

I have detailed what I mean by this in the attached file, I hope you find them useful. I enjoyed reading your paper. Best of luck with your research!

1.You may want to make sure that the abstract does not lean too much on text that is already in the paper. Sometimes it makes sense to express the same concepts differently.

We have changed the abstract due to our changes to the rest of the paper, and we hope that we did not lean too much on the text this time.

2.Your introduction reads well and I agree with you on the excessive price that diffusion models pay to epidemiology, especially the spread of diseases. In fact, most models of diffusion even use the same categories —e.g., immune, susceptible, infected — even though this may not have much sense in a social environment. I believe the broader category for these models is that of “threshold models of diffusion” (from Granovetter 1978 til Rosenkopf & Abrahamson 1999) and it would be a good idea for you to also acknowledge this literature more than you already do.

Granovetter M (1978) Threshold models of collective behavior. Am J Sociol 83(6):1420–1443.

Rosenkopf L, Abrahamson E (1999) Modeling reputational and informational influences in threshold models of bandwagon innovation diffusion. Comput Math Organiz Theory 5(4):361–384.

==== OUR RESPONSE ====

Thank you for these references. We added them into the introduction, as well as reference to the underlying sociological work.

==== REVIEWER ====

3.I have been working on diffusion model for quite some time now and I have found the literature review by Kiesling et al (2012) — a work that appears in your reference list — very much informative and well researched. They present and categorize diffusion models by considering what agent-based modeling brings to the theory. Even though they do not explicitly mention cognition, they do refer to individual characteristics (e.g., psychological, social) that affect the diffusion process. 

Moreover, there have been agent-based models (ABMs) of diffusion with a cognitive backbone (Secchi & Gullekson 2016). I am writing this because I do not think you can claim that you are introducing “a new class of models” (p.2, line 25). In fact, this class of agent-based models that focus on individual attitudes, dispositions, perceptions as a way to study diffusion are very much the reason why scientists have turned to agent-based models. Besides this point, it is my understanding that PLOS is not interested in novelty as such, scientific robustness would be enough to grant publication. In short, I do agree with your approach and I think it is sound but, at the same time, I do not see the need to oversell. This is both because what you claim as unique is part of the ABM approach and because PLOS journals are not the place where these claims make a difference.

Kiesling E, Günther M, Stummer C, Wakolbinger LM (2012) Agent-based simulation of innovation diffusion: a review. CEJOR 20(2):183–230.

Secchi D, Gullekson NL (2016) Individual and organizational conditions for the emergence and evolution of bandwagons. Comput Math Organiz Theory 22(1):88–133.

==== OUR RESPONSE ====

We very much value the clarification of our claims of novelty. We agree, given more background literature, that our cognitive function is not a novel contribution. We added text throughout the main paper to indicate that we rather view our contribution as an application of existing contagion techniques to the disinformation problem -- centering cascading contagion started by institutional agents. That reframing also culminated in changing the title and name of our model from a “cognitive contagion” model to a “cognitive cascade” model.

==== REVIEWER ====

4.Your reference for agent-based social systems [37] is not what I was expecting. I was almost certain to find one of the publications from Macal and North... I may be mistaken here.

==== OUR RESPONSE ====

We added in a reference to a paper we found helpful (Macal & North 2009) where we introduce agent-based social systems in the introduction:

Macal, Charles M., and Michael J. North. "Agent-based modeling and simulation." Proceedings of the 2009 Winter Simulation Conference (WSC). IEEE, 2009.

==== REVIEWER ====

5.Your call for simplicity in ABM (p.3, lines 67-75) is a bit puzzling, perhaps since you cite a cellular automata model (not an ABM) to make your point. There has been much discussion in the social simulation modeling community around descriptive vs simple models. What is now a widely recognized feature of ABM is the fact that they do not need to be limited to simple rules, agents, or environments. Quite the contrary, a strong point of using these models is the capacity to generate complex systems. It was Moss and Edmonds (2005) who first popularized the idea that one not only could but should exploit the more descriptive features of ABM. These ideas reflect on the various modeling purposes (Edmonds et al. 2019) in the sense that the intensity of description may depend on the general aim of the model. In short, I believe the picture is much more nuanced than what you hint at in the paragraph mentioned at the beginning of this point.

Edmonds B, Moss S (2005) From KISS to KIDS—an ‘anti-simplistic’ modelling approach. Lecture Notes in Artificial Intelligence. In: Davidson P (ed) Multi agent based simulation, vol 3415. Springer, New York, pp 130–144.

Edmonds B, Le Page C, Bithell M, Chattoe-Brown E, Grimm V, Meyer R, Montañola-Sales C, Ormerod P, Root H, Squazzoni F (2019) Different modelling purposes J Artif Soc Soc Simul 22(3):6.

==== OUR RESPONSE ====

We appreciate the references, but chose to keep our model aimed at simplicity. Because this argument regarding simplicity vs complexity is not crucial to the points made in our main paper, we removed the introduction paragraph advocating for it.

==== REVIEWER ====

6.You do a good job in succinctly explaining the logic of a simple diffusion model. At the same time, I was wondering why you have decided to assume that one of the two individuals in the example already holds a belief. This is a slightly more complicated case than the case where the recipient does not actually have prior beliefs. This does not mean that there is going to be a necessary ‘contagion’ but that the likelihood for that to happen is entirely based on a probability (the probability to ‘meet’ this belief) without any given prior. That is more suitable with the analogy of the disease, since one does not necessarily have antibodies (i.e. the prior), and the current pandemic is a sad reminder of this. Moreover, in diffusion of innovation studies — I keep mentioning it because this has been the benchmark for diffusion research for a long time — typically there is no prior, otherwise the innovation would not be innovative. In short, I think you should at least mention that you are keeping the exemplification closer to your approach and model rather than being closer to what most equation-based models have done in the past to cover simple diffusion.

==== OUR RESPONSE ====

Given this useful background on innovation diffusion practices, we added language addressing our departure from these conventions in the section on Simple Contagion: “As opposed to many studies in innovation diffusion (Zhang & Vorobeychik, 2019), we choose to argue in our formulation of contagion that every agent has a prior belief. Innovation diffusion studies often imagine any individual has no prior opinion about a new idea until they are ``infected'' with it. However, as we will illustrate below, we model beliefs on a spectrum, including belief in, against, and uncertainty about, a proposition. This departure from the epidemiological view of opinion diffusion allows us to argue for a prior, even if it is uncertainty.”

==== REVIEWER ====

7.There are two points that came to mind as I was reading on page 6:Page of 25

7.1. I follow your reasoning and it makes intuitive sense but I am not sure a model — especially after the claims you make on cognition — should be based on sole intuition. What is the theory behind the assumption that individuals would hold beliefs and be more or less prone to accept advice, recommendations, or information coming from others? This is a very important aspect that should motivate the way you model a computational simulation, especially when approaching ABM. Again, the point is not about the logic behind your modeling, but the justification of your assumptions. In my models, I use something very similar to this to justify individual dispositions towards others, and that is a version of Simon’s concept of docility (1993). I usually frame this as a behavioral aspect of distributed cognitive processes.Simon HA (1993) Altruism and economics. Am Econ Rev 83(2):156–161.

==== OUR RESPONSE ====

We hope that we have sufficiently motivated our DCC model from relevant literature on cognitive dissonance in relation to political or ideological beliefs (Van Bavel & Pereira, 2018; Bail et al., 2018; Swire & Lazar, 2020; Porot & Mandelbaum, 2020; Jost, 2009; Iyengar, 2009), and articulated that our model is similar to others who also model dissonance (Li et al., 2020; Goldberg & Stein, 2018; Baldassarri & Bearman, 2007).

==== REVIEWER ====

7.2. The other comment is more technical. At one point, you mention that you have used a way of coding strong/weak beliefs in a way that resembles a Likert scale. This is a justification that I have also used myself several times, especially at conferences to justify why certain variables were programmed the way they were. The problem I see with your way of interpreting this assertion is that you are being too literal. Psychometric scales are almost never measured on a single item, hence you never have discrete values, but a pseudo-continuous set of values for each individual that is derived from summating or averaging the score in the various items. Hence, either you are following standard measurement in cognitive psychology (using scales, not items) or you are doing something else (using discrete values). A minor point —irrelevant for the calculations — is that 7-point Likert scales are coded from ‘1’ to ‘7’ not from ‘0’ to ‘6’.

==== OUR RESPONSE ====

Thank you for this framing. In response to reviewer 2’s related comments, we conducted additional experiments using belief resolutions of 2, 3, 5, 9, 16, 32, and 64, whose results are included in the supplemental materials (S11 Text, S12-S21 Fig). We hope that these different belief scales break us away from overreliance on a 7-point Likert scale.

==== REVIEWER ====

8.As I keep reading the description of your model, I really cannot understand your claim of novelty as far as agents are kept holding heterogeneous beliefs (and dispositions) in an ABM. What you write seems standard bounded rationality assumptions for heterogeneous agents in a computational simulation model of a complex system. See the introduction of Edmonds E, Meyer R (2017). Simulating social complexity. A handbook. Springer.

==== OUR RESPONSE ====

We agree, and hope that our reframing of our contributions are now more in line with the extant literature.

==== REVIEWER ====

9.In your formula for a binary belief update function, you assume that the positive or negative distance from one’s own belief should be treated equally. However, we know that the sign of the distance is probably relevant, as one may infer from people’s understanding of losses and gains in famous experiments such as those by Kahneman and Tversky (1979). You may want to argue as of why it makes sense to take the absolute value of the difference in Eq. 4.

Kahneman D, Tversky A (1979) Prospect theory: an analysis of decision under risk. Econometrica 47(2):263–292.

==== OUR RESPONSE ====

In response to this and other literature recommended by reviewers that also utilized this notion (Dellaposta et al., 2013), we made a small addition to address this as we discuss the update function in Eq. 4 (displayed as Eq. 5 in the track-changes version): “There are similar functions motivated in contagion models centering dissonance (Li et al., 2020; Dandekar et al., 2013), and some which weight positive or negative influence differently (DellaPosta et al., 2016). We chose to weight positive or negative influence equally in this example, and subsequent contagion functions, to simplify the model and make its results more easily analyzable.”

==== REVIEWER ====

10.Eq. 5 is, as you explicitly state, very vague and there is a need to justify beta:

10.1. I am not sure why you have lost the ‘v’ in your reference belief, as if you are now treating the belief to which one is exposed as a-social or not coming from a source of any kind. Whether a belief spreads from a family member, a friend, social media, or the news, it still comes from someone. I am not sure why this turn to ‘objective’ belief as opposed to the one anchored to another agent —i.e. the model you have used thus far. Obviously, I would have some problems with the “objectivization” of beliefs.

==== OUR RESPONSE ====

Thank you for pointing this out. We updated equations (7) and (8) to include $b_v$ rather than just $b$.

==== REVIEWER ====

10.2. I have the impression that everything that has been discussed thus far is instrumental to presenting beta and Eq. 5. If this is the case, then you could probably re-organize the section and do it more straightforwardly, perhaps being more explicit about the quest for a beta function that captures cognition involved in misinformation spreading, to some extent.

==== OUR RESPONSE ====

With our new reorganization, we believe that as the cognitive contagion motivations are more contained in the background section, citing more works, we are less centering a build-up toward the beta function. We hope that now, our argument is more clearly aimed at the cascade model with a defensive contagion function and institutionally-driven cascades.

==== REVIEWER ====

11.From the visualization of your model, it seems that you have not created another “level” or layer (i.e. the institution) but another agent in the system. I mean, technically, this is what it is —that you have two different types of agents, one is to represent an individual (cognizer) and the other is there to represent a news outlet (an institution). Of course, you can interpret this to be two different layers but the programming does not necessarily indicate that this is the case. This means that a modeler reading your article would need a more detailed description as of why this two-agent strategy is actually identifying two layers. I do not think this is too difficult, as I have done it in some of my models—it just needs a more substantial justification.

12.As I read this section, I have noticed that you state that ABM do not often model “levels”. I believe this depends on which field you are looking at. For models in economics, for example, it is customary to model a market and its agents. For models in organizational research, this is also something that happens quite often. This is related to the ability that ABM has to allow for the modeling of fast and slow timescales or, at least, this is the explanation that I think is the most valuable. You could have a look at Neumann M, Cowley SJ (2016). Modeling social agency using diachronic cognition: learning from the Mafia. In Agent-Based Simulation of Organizational Behavior (pp. 289-310). Springer, Cham.

==== OUR RESPONSE ====

We revisited the work we are citing to more clearly make the argument we attempted. Instead of claiming that ABM do not model levels, we should have claimed that they _do_, but we are trying to get away from the typical way they are modeled. We added text in the first two paragraphs of the “Institutional Cascades” subsection to clarify.

==== REVIEWER ====

13.The presentation of the model does not seem to follow any of the known protocols or schemes that are found in specialized journals for ABM such as, for example, the Journal of Artificial Societies and Social Simulation (JASSS). I am referring to the good custom to show, for example, a flow chart summarizing the processes, and to split the description of agents and their characteristics from the processes of the model. You may want to add a table where you define each parameter, their notation, values, experimental values, and a short description. While published articles do not include the entire ODD protocol (see Grimm et all 2020 for the latest version), it is good to include some of it.

Grimm V, Railsback SF, Vincenot CE, Berger U, Gallagher C, DeAngelis DL, Edmonds B, Ge J, Giske J, Groeneveld J, Johnston AS (2020). The ODD protocol for describing agent-based and other simulation models: A second update to improve clarity, replication, and structural realism. Journal of Artificial Societies and Social Simulation 23(2):7.

==== OUR RESPONSE ====

We agree with this suggestion, and have added a table under the section “Comparing Contagion Methods” which details our parameter selection.

==== REVIEWER ====

14.One of the aspects that characterizes ABM research is that these simulations lean (more or less) heavily on stochastic components. From your presentation of the model, I am not sure I have a clear understanding of where exactly are the stochastic components of the model. Of course, a limited part of these components suggests that, perhaps, ABM may not be the preferred choice for a computational simulation.

==== OUR RESPONSE ====

The stochastic elements are in the distribution of initial beliefs, in the probabilities guiding the spread of beliefs, and in the network topology, randomly constructed for each simulation.

==== REVIEWER ====

15.The three conditions (p.9) you describe for the institutional distribution of messages are interesting and make sense from a computational/technical point of view, because you want to understand the impact of a variety of messages. At the same time, they should reflect conditions that can be observed in an actual institutional environment, unless the simulation wants to be a theoretical (in principle) case. Given the actuality of your topic, I do not think this latter is the case and so, I would ask if you could provide some example of what these conditions refer to.Page of 45

==== OUR RESPONSE ====

In response to similar questions by reviewer 1, we added language in the Future Work section to address the possibility of more realistically modeling the media ecosystem and message sets.

==== REVIEWER ====

16.I am not sure it is standard practice in ABM to write hypotheses (p.10) as a way to present the expectations on what to find. However, if you must, then you should at least (a) ground them in the literature and (b) explicitly formulate hypotheses. I would prefer that you are not explicit on hypotheses since, if that is the case, then formal testing needs to be performed and that would probably take the manuscript a bit far from what you intend to do.

==== OUR RESPONSE ====

Though we appreciate the comment, we chose to leave the language as is. We hope that our use of “hypothesis” helps to clarify our thought process in conducting the experiments.

==== REVIEWER ====

17.When you write about “preliminary experiments” (p.10, line 341) to find the values of the threshold and the contagion probability, what is it exactly that you did? Did you perform a sensitivity analysis? And how come a constant is the outcome of such an analysis. ABM usually provide a range of values for their parameters.

==== OUR RESPONSE ====

We now detail the process by which we chose our simple and proportional threshold parameters in the supplemental material (S1 Text & S2 Table). We hope that this helps to explain our decisions.

==== REVIEWER ====

18.If I understand what you write about the model and your intention is to test the effect of different graph types, I am not sure why you have decided to go with ABM as opposed to a more classic social network analysis. This point, in connection with the one on stochasticity above, needs further (better) explanations.

==== OUR RESPONSE ====

We believe that the language that we added concerning our contribution being a cascade model that leverages both network science techniques and ABM techniques helps explain our work. We hope that our work borrowing from both disciplines allows both to benefit from the strengths of each other.

==== REVIEWER ====

19.Also, the determination of the number of runs for each configuration of parameters seem very vague. You write you set on 10 without providing any detail about what procedure you followed. This is a very important aspect to understand whether results are valid.

==== OUR RESPONSE ====

We agree, as this point also mirrors one made by reviewer 2. To address this, we chose to aggregate results for 10, 50, and 100 simulation runs for each configuration, and do correlation analyses to determine whether the results differed significantly. Results are in the supplemental material (S22 Text, S23 Table, and S24 Fig), and we use them to justify using 10 runs in the main paper.

==== REVIEWER ====

20.Results are overall well presented and described. I have only two comments:

20.1. The first impression is that you are trying to do too much with just one paper. There are many conditions that you want to test with this simulation and results are rich. However, too much information is probably as dangerous as its paucity. Maybe there is a way to limit the amount of graphs and comments and this could be done with a more structured systematic approach to the findings. You can start by presenting a table where computational experiments are classified and indicate the main finding for each one of them. Some results are similar and you could skip them, unless significant differences appear in the configuration tested. You may want to merge the presentation of results and their analysis, where some of these summary tables already appear. Given these tables, I am not sure whether the many graphs provide additional information.

20.2. The second comment is about the uncharacteristic form that most of your graphs show in relation to the type of simulation modeling you have conducted (ABM). This points at the minimal stochasticity that seems to be embedded in your model. I may be too much leaning towards models with a large stochastic component, but linear results such as the ones that most of your results show require an explanation or, maybe, just a more straightforward presentation of the ABM.

==== OUR RESPONSE ====

We hope that we have now sufficiently articulated how our model does have stochastic elements, and that some of the supplemental materials (S3 & S4 Fig) demonstrate the significant variation in results for some experimental combinations.

==== REVIEWER ====

21.I think I can stop here with my comments and come back to the final part of the paper once you have addressed the above

---

## [Decision Letter · Decision Letter 1]

19 Oct 2021

PONE-D-21-06938R1Cognitive cascades: How to model (and potentially counter) the spread of fake newsPLOS ONE

Dear Dr. Rabb,

Thank you for submitting your manuscript to PLOS ONE. The review process has taken longer that it usually does, but the reviewers have been accurate and overall they clearly appreciated the improvements following the first version of the manuscript. The work is now close to fully meet publication criteria, with some aspects that still require additional consideration. However, the Minor Revision status does not imply that reviewers' last comments could be considered lightly. They are comments that could further improve the manuscript's quality, so appropriate consideration is required.Therefore, we invite you to submit a revised version of the manuscript that addresses the points raised during the review process. In particular:Reviewer 2 pointed to a single issue regarding tests robustness and the validity of conclusions, which seems to need further explanations to be completely convincing. This is an important point.Reviewer 3, instead, suggests a list of minor modifications or in some cases to add in the manuscript some explanations that were only given in the answers to reviewers. 

We look forward to receiving your revised manuscript.

Kind regards,

Marco Cremonini, Ph.D.

Academic Editor

PLOS ONE

Journal Requirements:

Reviewers' comments:

Reviewer's Responses to Questions

**Comments to the Author**

1. If the authors have adequately addressed your comments raised in a previous round of review and you feel that this manuscript is now acceptable for publication, you may indicate that here to bypass the “Comments to the Author” section, enter your conflict of interest statement in the “Confidential to Editor” section, and submit your "Accept" recommendation.

Reviewer #1: All comments have been addressed

Reviewer #2: (No Response)

Reviewer #3: (No Response)

2. Is the manuscript technically sound, and do the data support the conclusions?

Reviewer #1: Yes

Reviewer #2: Partly

Reviewer #3: Yes

3. Has the statistical analysis been performed appropriately and rigorously? 

Reviewer #1: Yes

Reviewer #2: Yes

Reviewer #3: Yes

4. Have the authors made all data underlying the findings in their manuscript fully available?

Reviewer #1: Yes

Reviewer #2: Yes

Reviewer #3: Yes

5. Is the manuscript presented in an intelligible fashion and written in standard English?

Reviewer #1: Yes

Reviewer #2: Yes

Reviewer #3: Yes

6. Review Comments to the Author

Reviewer #1: (No Response)

Reviewer #2: Thank you for responding so thoroughly to my previous comments. The only area of outstanding concern for me is in the sensitivity test to the number of levels in your model. In my previous review, I asked that you check that the conclusions of your model are not sensitive to the arbitrary number of levels you chose. In theory, if the results are rigorous they should hold equally well despite the number of levels. Thank you for conducting this sensitivity test and reporting the results.

I am worried however that your tests did not find that your conclusions were truly robust to the number of levels in the model. In your response and supplement you state this, but in the paper itself you merely say that you ran a sensitivity test. The main body of the paper makes no mention of the fact that the results seem to be strongly influenced by the discretization assumption. In fact, the sentence "We additionally ran in-silico experiments with lower and higher resolutions..." seems to imply that you didn't see any difference by going to more continuous levels of belief, especially as you had previously said "bu can be a continuous variable with the interval from strong disbelief to strong belief, or it can take on discrete values". At the very minimum, you have a disconnect between the scope condition you are claiming (continuous levels belief) and the domain over which your model predicts the outcome you claim (discrete belief levels). To be completely honest with your readers, I think more of this discussion of the reliance on discrete belief levels belongs in your main text. Don't let anyone suspect you're hiding anything.

What your discretization assumption seems to be doing is acting as a coarse proxy for similarity between neighbors, so that you don't have to justify a rule for who individuals pay attention to that works in the continuous domain. As your results are dependent on this, you need to be more explicit about it. Otherwise, you can update your model to allow for similarity given a continuous measure of belief, and see if your results still hold. This would be a more robust result, and easier for you to justify if it doesn't add too much modeling complexity.

Thanks,

James

Reviewer #3: My report is in the file attached to this form. Please open that file to access my comments to the paper.

7. PLOS authors have the option to publish the peer review history of their article (what does this mean?). If published, this will include your full peer review and any attached files.

Reviewer #1: **Yes: **Giacomo Livan

Reviewer #2: **Yes: **James Houghton

Reviewer #3: **Yes: **Davide Secchi

---

## [Author Response · Author response to Decision Letter 1]

3 Dec 2021

We would like to thank all reviewers for their comments and suggestions, which we believe again improved our revised draft of the paper. Specific responses to individual questions from reviewers can be found below.

REVIEWER 1

N/A

REVIEWER 2

Reviewer 2: Thank you for responding so thoroughly to my previous comments. The only area of outstanding concern for me is in the sensitivity test to the number of levels in your model. In my previous review, I asked that you check that the conclusions of your model are not sensitive to the arbitrary number of levels you chose. In theory, if the results are rigorous they should hold equally well despite the number of levels. Thank you for conducting this sensitivity test and reporting the results.

I am worried however that your tests did not find that your conclusions were truly robust to the number of levels in the model. In your response and supplement you state this, but in the paper itself you merely say that you ran a sensitivity test. The main body of the paper makes no mention of the fact that the results seem to be strongly influenced by the discretization assumption. In fact, the sentence "We additionally ran in-silico experiments with lower and higher resolutions..." seems to imply that you didn't see any difference by going to more continuous levels of belief, especially as you had previously said "bu can be a continuous variable with the interval from strong disbelief to strong belief, or it can take on discrete values". At the very minimum, you have a disconnect between the scope condition you are claiming (continuous levels belief) and the domain over which your model predicts the outcome you claim (discrete belief levels). To be completely honest with your readers, I think more of this discussion of the reliance on discrete belief levels belongs in your main text. Don't let anyone suspect you're hiding anything.

==== OUR RESPONSE ====

Thank you for this point, we agree. We have added language to clarify that while we can model things that approach continuity, that for some of our experiments, the dynamics became different when the number of different “belief levels” became large enough (while still showing there is nothing magic about 7 and that slightly fewer or more but still discrete belief states gave similar network dynamics. ) We suspect this is a limit of the way we set initial model parameters during experiments, more than something intrinsically different about true continuous belief states, but we now make it very clear that there’s a difference in the main paper.

==== REVIEWER ====

What your discretization assumption seems to be doing is acting as a coarse proxy for similarity between neighbors, so that you don't have to justify a rule for who individuals pay attention to that works in the continuous domain. As your results are dependent on this, you need to be more explicit about it. Otherwise, you can update your model to allow for similarity given a continuous measure of belief, and see if your results still hold. This would be a more robust result, and easier for you to justify if it doesn't add too much modeling complexity.

==== OUR RESPONSE ====

Completely agree – see above. 

REVIEWER 3

Reviewer 3: Thank you very much for your hard work on the manuscript, I think the revision has improved it significantly from the previous version. I have enjoyed reading this version as well as the previous one and I believe it makes an important contribution to the literature. As an agent-based modeler, I am still slightly uncomfortable with some of your choices concerning modeling and their reporting on the paper. However, most of my concerns have been addressed and I am more comfortable with this version of the paper. There are a few open points that you may want to address before submitting the paper in its final version. I have reported them below —some refer to previous comments that require additional attention while others are new. I assess these comments as requiring minor efforts on your part.

31.Former point 3. I think your revised introduction clarifies the scope and objectives of the paper well —much better than before. One point that I think still requires your attention is the definition of a cognitive ‘cascade’. In fact, I cannot find a place where you define what you mean with this. You simply write that cognitive cascade models are “those that adopt a cascading network-based diffusion model from social network theory” (p.2, lines 26-27). Fine, but what is the cascade in a cascading network-based model? It seems you define cascading models through cascading networks. The result is a very unclear definition to those who do not know the meaning of a cascade.

==== OUR RESPONSE ====

We now define “cascades” explicitly in the introduction, and contrast it to how we intend to use the word “contagion.”

==== REVIEWER ====

32.Former point 5. When you refer to ABM in the social sciences you still refer to Schelling’s cellular automata model [48]. My comment was probably unclear on this point, since I was also trying to make another point (which you addressed). Apologies for repeating but [48] is not an ABM. You can cite any ABM published in JASSS to make your point or refer to the classic books by Gilbert, Edmonds, Troitzsch, etc. Here are some of them:

Edmonds E, Meyer R (2017). Simulating social complexity. A handbook. Springer.

Gilbert, N. (2019).Agent-based models(Vol. 153). Sage Publications.

Gilbert, N., & Troitzsch, K. (2005).Simulation for the social scientist. McGraw-Hill Education (UK).

==== OUR RESPONSE ====

We have now changed out the Schelling 1971 reference to the two references you provided from Gilbert.

==== REVIEWER ====

33.Former point 7.2. I have mentioned the use of Likert scale before and the fact that I find this approach very much in line with how modelers reason about numerical representations of beliefs. I do not quite understand your reply though. To my claim that you have been too literal and used a discrete value approach rather than mimicking the use of a Likert scale you have replied by adding a sensitivity analysis of the values of parameter b. I understand you performed those checks as reply to a point from another reviewer but that does not address my point. Please do let me know if I am missing something here. Do not get me wrong, I do think that your approach is fine; it is just that your claim is not fully consistent with the use of a Likert scale in applied psychology.

==== OUR RESPONSE ====

By showing that the model is somewhat robust to whether a 7- or 5- or 9-point scale is used (where we indeed used 7 as inspired by Likert scales), we hoped that this would dissuade any interpretation of our choice of 7 points as trying to exactly emulate Likert scales. Hopefully we have removed any indication that we are exactly using the Likert scale the way it is used in applied psychology. 

==== REVIEWER ====

34.Former point 9. I could not find your response in the text of the article. I think it is a convincing one, please go ahead and find room for it.

==== OUR RESPONSE ====

We wrote language after Eq. 4 to capture what we wrote in the previous comments.

==== REVIEWER ====

35.Former point 14. I appreciate your reply and it may be a good idea to specify these points somewhere in the paper

==== OUR RESPONSE ====

Thank you. We captured this now at the end of the “Experimental Design” subsection.

==== REVIEWER ====

36. Former point 16. I respect your choice of leaving hypotheses but then I would expect them being accepted or rejected following some more formal testing procedure. Another way to do this would be to call them differently (e.g., assumptions, propositions) so that a reader does not expect formal testing. Now, the test you have used is correlation (I could see Chi-square tests from some of your tables as well). You may want to use that to test your hypotheses although it is a test of association, not one that would help you understanding whether the spread (or lack thereof) was determined to some extent by the conditions of the simulation. Maybe a test where you can actually show dependence between conditions and output would probably make your point stronger.

==== OUR RESPONSE ====

In order to be less confusing, we have changed the word “hypothesis” or “hypothesize” to “we expect” or “we anticipate” that do not imply that we will conduct statistical tests to test those hypotheses.

==== REVIEWER ====

37.Former point 19. I have read your supplementary materials files and I am afraid I was not able to find a justification for the 10 repetitions. In one of these files you state “We varied p, the contagion probability, from 0.05 to 0.95 in increments of 0.05, and took averages over 10 simulation runs”. This means that 10 was a given. “Why 10” and not 9 or 13, 57 or 9762 was my question.

==== OUR RESPONSE ====

Thank you for catching this. We agree that our previous answer did not address that aspect. We have run additional correlation tests and included a write-up of results in supplementary material S1 Text.

==== REVIEWER ====

38.Former point 20.1. I still think that you can find a better —more succinct way —of presenting your results. There are currently 14 figures in the paper and these are probably too many. Unless you convincingly argue that they all have something unique to tell about your results, I would consider moving some to the supplementary materials. Another way to deal with this is that of presenting results together with their analysis. You do not need to describe and then analyze, you can describe as you analyze results. I appreciate you do the analysis a bit more precisely, but some graphs can be made to compare results already and differences could be shown as a comment to those graphs.

Is there something we can move to the supplement? Doesn’t hurt. 

==== OUR RESPONSE ====

We agree that this was a verbose way to present our results. During our presentation of results under the section “Comparing Contagion Methods,” we have removed some of the superfluous description of results, as well as moved the simple and proportional threshold cascade results to the supplements.

==== REVIEWER ====

39.A minor point: In the introduction you write about social network theory and cognition theory. I believe you want to refer to social network research (or analysis) and cognitive science. In particular, the latter (i.e. “cognition theory”) does not exist although there are many cognition theories that belong to the different domains of cognitive science. I suggest you make the reference to theories in the plural or change them accordingly.

==== OUR RESPONSE ====

We have changed “cognitive theory” to simply say “cognitive science” instead to avoid confusion.

==== REVIEWER ====

40.If you are mainly referring to the status of misinformation in the US, you should explicitly state it at the beginning of the article, when you bring in the first example. If you do not do that, the problem is that you assume readers implicitly think of the context as the US when, in reality, an article can be read from anywhere in the world. Maybe using the US as a reference has its advantages in the sense that misinformation seems to have reached particularly concerning levels there as compared to other countries. [I am using vaccination rates as a proxy and numbers are alarming if one compares the US with, say, Spain or the European Union as a whole.] So, please, try to be less US-centric and inform readers that you are taking this perspective.

==== OUR RESPONSE ====

This is a welcome point, thank you for bringing it to our attention. We have added explanations that we are primarily focusing on evidence from the U.S., even though COVID misinformation appears in many countries. We now make this more clear in the introduction.

==== REVIEWER ====

41.I appreciate the new focus on cognitive cascade models although you keep swinging between contagion and cascade (pp.5f). It may be a good idea to clarify the connection between the two and, at the same time, try and be more consistent in the use of terminology throughout the paper.

==== OUR RESPONSE ====

Thank you for catching the inconsistencies. We now consistently use the language of “cascades” where applicable, and when mentioning “contagion,” make clear the difference, as we define it, between individual-level contagion and macro-level cascades.

==== REVIEWER ====

42.Continuing from the point above, at page 6 you give the impression that the cascade is an event (“during a cascade”, line 196). This reminds me of the fact that you still have not defined what a cascade is. Something close to a definition comes much later (p.8).

==== OUR RESPONSE ====

As with the above point 31, we now define cascades explicitly in the introduction and differentiate it from contagion.

---

## [Editor Report · Decision Letter 2]

13 Dec 2021

Cognitive cascades: How to model (and potentially counter) the spread of fake news

PONE-D-21-06938R2

Dear Dr. Rabb,

We’re pleased to inform you that your manuscript has been judged scientifically suitable for publication and will be formally accepted for publication once it meets all outstanding technical requirements.

Kind regards,

Marco Cremonini, Ph.D.

University of Milan

Academic Editor

PLOS ONE
---

## [Editor Report · Acceptance letter]

28 Dec 2021

PONE-D-21-06938R2 

Cognitive cascades: How to model (and potentially counter) the spread of fake news 

Dear Dr. Rabb:

I'm pleased to inform you that your manuscript has been deemed suitable for publication in PLOS ONE. Congratulations! Your manuscript is now with our production department. 

Kind regards, 

on behalf of

Dr. Marco Cremonini 

Academic Editor

PLOS ONE